# KnowData: Knowledge-Enabled Data Generation for Improving Multimodal Models

## Abstract

In this paper, we introduce a novel framework to enhance the quality of synthetic image-text pairs for multimodal models such as CLIP. Our approach, named KnowData, integrates real-world knowledge explicitly into the generation of text descriptions. It combines structured knowledge from knowledge graphs like ConceptNet and unstructured knowledge extracted from Wikipedia, to ensure that the generated text descriptions are both contextually rich and accurately reflective of real-world knowledge. Additionally, we leverage Large Language Models for the expansion, summarization, and refinement of the text descriptions to ensure their coherence. These enriched texts are subsequently used to generate images through advanced text-to-image models like Stable Diffusion and DALLE-3. CLIP models are then fine-tuned with these synthetic data for downstream zero-shot image classification tasks. Our experiments across 9 datasets demonstrate that CLIP models fine-tuned with our knowledge-guided synthetic datasets outperform 6 state-of-the-art zero-shot CLIP methods (e.g., +11.23% on DTD and +4% on EuroSAT based on ViT-B/16 model; +11.47% on CIFAR-100 and +7.99% on DTD based on ResNet-50 model). These results showcase the improved out-of-distribution robustness and adaptability of KnowData across a diverse set of data domains. We further verify the design of KnowData through ablation studies, revealing that the integration of knowledge in the text descriptions contributes to the reliability, diversity, and detail orientation of the synthetic images, thereby offering better data scaling laws for CLIP zero-shot image classification performance.

## 1 Introduction

Multimodal learning, particularly in image-text models such as Contrastive Language-Image Pre-training (CLIP) (Radford et al., 2021) has witnessed transformative advancements in recent years. These models excel in understanding and correlating the nuances of visual and textual data, leading to applications in a wide range of domains. Despite their versatility, a critical aspect of their development hinges on the quality and relevance of their training datasets. Traditional dataset collection methods, predominantly based on extensive web crawling, could compromise on the contextual richness and accuracy of text-image pairs due to the inclusion of noisy data on the internet (Feng et al., 2024).

To fill in this gap, there have been several approaches proposed to improve the quality of image-text pairs as training or fine-tuning data. For instance, to enhance data quality when building the powerful DALLE-3 model, Betker et al. (2023) trained a bespoke image captioner to recaption the image dataset. Also, it has been shown that merely enlarging the training data size or combining multiple sources does not necessarily lead to better multimodal models, while the data quality plays the key role (Nguyen et al., 2022; Fang et al., 2022). Existing studies mainly focus on using the implicit knowledge in language models to improve the text quality of image-text pairs (He et al., 2023; Shipard et al., 2023), which may lack factuality and diversity (Betker et al., 2023). In our work, we explore the question: *Can we explicitly integrate real-world knowledge to improve the quality of image-text pairs and further improve the performance and robustness of multimodal models?*

To explicitly leverage real-world knowledge to improve data quality, we propose a novel data generation framework KnowData (shown in Figure 1), which introduces a knowledge-guided approach to generate text-image pairs with multiple knowledge sources, including large-scale knowledge graphs,

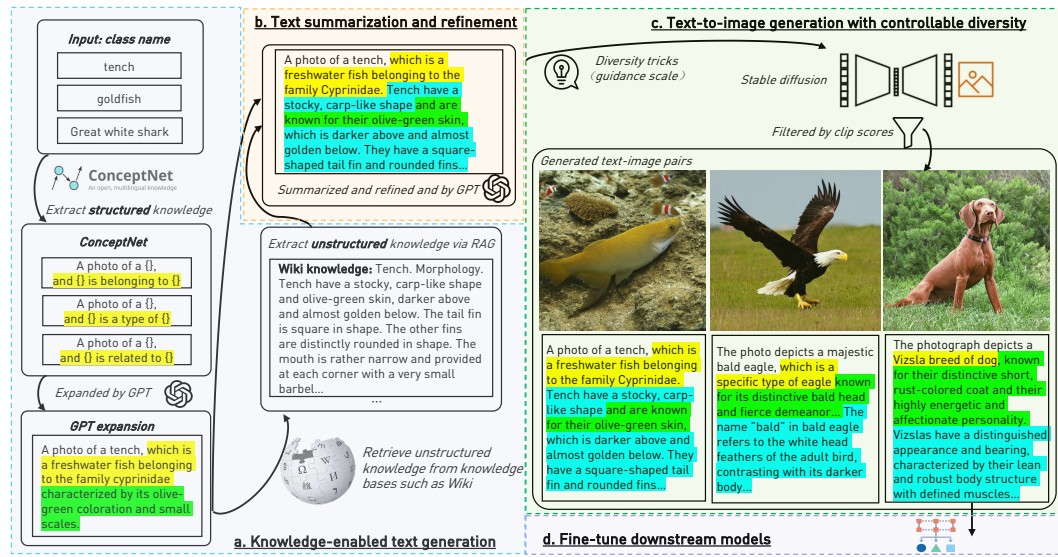

Figure 1: The proposed KnowData framework consists of four components: a) knowledge-enabled description generation, b) description summarization and refinement, c) image generation with controlled diversity, and d) downstream model fine-tuning. Here yellow texts indicate structured knowledge added through the knowledge graph (e.g., ConceptNet), blue texts signifies unstructured knowledge augmented from knowledge stores (e.g., Wikipedia) through RAG, and green texts represent knowledge expanded through LLMs.

Wikipedia knowledge stores, and Large Language Models (LLMs). In particular, **(1)** we first leverage the *structured* knowledge from knowledge graphs such as ConceptNet (Speer et al., 2017) to generate text that explicitly reflects basic object properties and relations (e.g.,"*Vizsla is related to dog*"). **(2)** Furthermore, we use LLM to *expand* the structured knowledge sentences for more coherent descriptions with supplementary details. **(3)** We then integrate the *unstructured* knowledge extracted from external knowledge stores. We build a Retrieval Augmented Generation (RAG) pipeline to extract related knowledge and description from Wikipedia (e.g., "*The Hungarian Vizsla is a short-coated hunting dog...The nose of the Vizsla will always have a reddish color that blends with the coat color.*"). **(4)** We then pass the text generated based on structured and unstructured knowledge to LLM to *refine and summarize* it, which leads to texts following the real-world text data distribution while containing multi-source knowledge. **(5)** Next, we use text-to-image models like Stable Diffusion (Rombach et al., 2022) or DALLE-3 (Betker et al., 2023) to generate images based on the refined texts. For each text input, we integrate different diversity constraints to generate multiple images, and select the high-quality pairs based on certain criteria such as CLIP scores (Radford et al., 2021). **(6)** Finally, we leverage our generated data to fine-tune the multimodal models. By integrating contextual and domain-specific insights into the text generation process and diversity tricks into the image generation phase, we aim to create high-quality training datasets, thereby enhancing the learning efficacy and application potential of multimodal models like CLIP.

Our extensive experiments demonstrate that CLIP models fine-tuned with our knowledge-guided, synthesized dataset outperform those trained with state-of-the-art (SOTA) data generation approaches (He et al., 2023; Shipard et al., 2023) and other zero-shot techniques (Allingham et al., 2023; Menon & Vondrick, 2023; Ge et al., 2023). We systematically evaluate KnowData across 9 datasets, highlighting its robustness and adaptability in various data domains. For instance, on ViT-B, we achieve performance improvements of 11.23% on the DTD dataset and 4% on the EuroSAT dataset compared to the SOTA. On RN50, we achieved improvements of 11.47% on the CIFAR-100 and 7.99% on DTD. Furthermore, our ablation studies show that (1) by gradually adding more knowledge sources for text descriptions, KnowData produces synthetic images with better reliability, diversity, and accurate details; (2) KnowData benefits from stronger text-to-image generators; (3) KnowData enables better data scaling law; (4) diversity in text knowledge and diversity in images both matter.

Our findings highlight the effectiveness of knowledge-infused synthetic data in enhancing CLIP models' generalization capabilities, suggesting a need to reevaluate dataset design strategies in

multimodal learning. This study aims to present a new data generation pipeline and ignite further research into knowledge-guided approaches for multimodal learning.

## 2 RELATED WORK

To demonstrate the capability of our knowledge-enabled data generation, we primarily evaluate our generated data on improving CLIP zero-shot classification performance (i.e., without using real data). Existing approaches to improving CLIP's zero-shot performance include *enhancing the per-class text embedding with additional content*, and *fine-tuning CLIP model with synthetic data*.

**Enhancing the per-class text embedding with additional content.** For vision-language models like CLIP, the classification outcome is determined by finding the class label whose text embedding is most similar to the image embedding. Enhancing text descriptions about class labels can improve CLIP classification performance. For example, Allingham et al. (2023) use LLMs to generate additional text prompt templates and perform weighted selection among these templates, while Menon & Vondrick (2023) uses LLMs to derive descriptions for class names to obtain text embeddings. Ge et al. (2023) supplements class names with hierarchical knowledge from WordNet to enhance text embeddings. Despite leveraging knowledge, these methods lack systematic, explicit injection of diverse and accurate knowledge, which may result in irrelevant or false information generated from LLMs. Our approach uses multiple knowledge sources to ensure both diversity and accuracy. Moreover, the aforementioned methods are limited to manual adjustments at the text embedding level. In contrast, we use knowledge to generate synthetic text-image pairs and then fine-tune the model, which can provide more flexible and thorough model adjustment.

**Fine-tuning models with synthetic data.** Existing studies employ various generation tricks for diffusion models to enhance the diversity and quality of generated images and then finetune the CLIP model, but they fail to effectively incorporate the relevant knowledge of the class itself into the generation process (Shipard et al., 2023; Sarıyıldız et al., 2023). He et al. (2023) employs a word-to-sentence T5 model to enrich prompts and generate images, but this approach merely randomizes class expressions without systematically enriching class-related knowledge. Other studies, such as Bansal & Grover (2023) and Trabucco et al. (2023), use synthetic data for data augmentation, and Fan et al. (2023) explore the scaling laws of synthetic images for image classification tasks, without enhancing the quality of generated data. In contrast, our knowledge-enabled data generation framework produces higher-quality text descriptions and synthetic images with more detailed features, enhancing CLIP zero-shot performance on downstream tasks.

**Text-to-Image Diffusion Models.** Diffusion models have significantly advanced text-to-image generation by producing high-quality images from textual descriptions (Sohl-Dickstein et al., 2015; Ho et al., 2020; Nichol & Dhariwal, 2021). Notable models like Stable Diffusion (Rombach et al., 2022), Imagen (Saharia et al., 2022), GLIDE (Nichol & Dhariwal, 2021), and DALLE-3 (Betker et al., 2023) have demonstrated impressive capabilities in this domain. However, these models often lack explicit knowledge utilization, leading to synthetic images that may miss certain detailed features. This gap motivates our exploration of knowledge-enabled synthetic data generation.

## 3 KNOWDATA

In this section, we describe how we generate knowledge-enabled texts (Section 3.1 and Section 3.2), create diverse images based on these texts by adding image diversity constraints (Section 3.3), and utilize the synthetic data to fine-tune the downstream models (Section 3.4).

### 3.1 KNOWLEDGE-ENABLED DESCRIPTION GENERATION

Our knowledge-enabled description generation pipeline is designed to produce high-quality prompts from a given class name. Subsequently, these prompts can be utilized by text-to-image diffusion models to generate superior images. Our pipeline integrates diverse types of knowledge from various sources. Specifically, we incorporate structured knowledge derived from large-scale knowledge graphs, and unstructured knowledge from external knowledge stores via the Retrieval Augmented Generation (RAG) framework and LLM, thereby enriching the details of the generated text.

Formally, let $\{c_1, \ldots, c_K\}$ be a list of targeted classes names, where $K$ is the total number of classes. We generally use $x$ to denote the class description prompt used for image generation within the

pipeline. The naive base prompt $x_i^{base}$ can be "*A photo of $c_i$*" for each class $c_i$, where $i \in [K]$. Next, we will elaborate on how we improve over the base prompt by incorporating diverse knowledge.

**Extracting unstructured knowledge from knowledge graphs.** Knowledge graphs are advanced data structures that map out the connections between various entities, such as objects, people, places, and concepts, to organize and integrate structured knowledge from multiple sources (Hogan et al., 2021). Considering their comprehensive and interconnected representation of information, we first supplement the classes with commonsense knowledge based on external knowledge graphs. The use of external knowledge graphs does not rely on additional models and, most importantly, ensures the *correctness and broadness* of the integrated knowledge. In this work, we choose ConceptNet (Speer et al., 2017) for structured knowledge extraction. Unlike some other knowledge graphs, such as ATOMIC (Hwang et al., 2021) that provides commonsense knowledge around human events, ConceptNet is more focused on encyclopedic knowledge. This aligns well with the natural image datasets we plan to evaluate, such as ImageNet.

Specifically, each node in ConceptNet represents one entity (e.g., object), and the edge represents the relations between entities. We query the ConceptNet API with each class name $c_i$ as input, which will return triplets of {head, relation, tail} where $c_i$ appears either in the head or tail. We consider 18 relations that may benefit our image recognition task, such as "*RelatedTo*", "*IsA*", "*PartOf*", "*LocatedNear*", and then only select triplets describing those relations. We then create templates to convert these relations into more understandable sentences, e.g., replacing "*{} RelatedTo {}*" with "*{} is related to {}*". We defer the complete list of 18 chosen relations and corresponding alternative templates to Appendix C. Then we concatenate the base prompt $x_i^{base}$ with the ConceptNet sentence (e.g., "*A photo of $c_i$, and $c_i$ is related to {}*") to obtain our description in the knowledge graph enhancement stage of KnowData. We retrieve $N$ such structured knowledge descriptions for each class $c_i$, denoted as $x_{i,j}^{kg} = \mathrm{KG}_j(x_i^{base})$ where KG denotes the knowledge graph, and $j = 1, \ldots, N$.

**Enhancing commonsense knowledge rule with LLM.** After obtaining the basic commonsense knowledge, we use GPT-3.5 (Ouyang et al., 2022) to introduce rich context descriptions. This is because even after introducing related entities via ConceptNet relations, these commonsense knowledge descriptions based on our templates remain too brief. The expansion by GPT-3.5 allows for an enhanced expression of this common sense knowledge with higher quality vocabulary, syntax, semantic coherence, etc. Furthermore, as GPT-3.5 is a model pretrained with a vast amount of knowledge, it can also further supplement the knowledge in its generation, leading to descriptions with richer details (see examples in Figure 1 green texts).

Concretely, denote the LLM (e.g., GPT-3.5) as L. For each ConceptNet knowledge $x_{i,j}^{kg}, j \in [N]$ in each class $c_i$, we prompt LLM with "*Rewrite the sentence to make the description more detailed: {$x_{i,j}^{kg}$}*" to expand and supplement the sentences. We obtain $x_{i,j}^l = \mathrm{L}(x_{i,j}^{kg})$ as LLM expansion output.

**Retrieval Augmented Generation based on Wikipedia.** We find that LLM-enhanced descriptions still lack sufficient details about the class object and could contain hallucinated content. Therefore, we utilize Retrieval Augmented Generation based on Wikipedia (Wikipedia, 2004), a reliable knowledge store commonly used to fench factual knowledge, so as to add sufficient details about features of the object. For instance, the pure class name "*tench*" lacks descriptions of its physical features, while the explicit knowledge from Wikipedia can supplement it (see examples in Figure 1 blue texts).

In particular, we employ ColBERT (Khattab & Zaharia, 2020) for retrieval from a text corpus based on their pre-built Wikipedia index and obtain related information given the query (Semnani et al., 2023). To select related passages, a retrieval model employs an encoder function that projects texts into an embedding space, and then identifies passages that closely resemble the query instance. In essence, the retrieval function assesses the similarity between two textual instances within this embedding space. Following this, a K-Nearest Neighbors (KNN) approach is utilized to identify the most similar passages with high embedding similarity. Formally, let $\mathrm{RAG}$ be the retrieval model. For each LLM expanded description $x_{i,j}^l$ for each class $c_i$, we retrieve top $N_{rag}$ relevant passages from Wikipedia knowledge store through RAG, which are denoted as $p_{i,j,k}^{rag} = \mathrm{RAG}_k(x_{i,j}^l), k = 1, ..., N_{rag}$.

## 3.2 DESCRIPTION SUMMARIZATION AND REFINEMENT

For each class $c_i$, given each LLM expanded commonsense description $\{x_{i,j}^l\}$ and each relevant detailed knowledge description $\{p_{i,j,k}^{rag}\}$ retrieved from Wikipedia, we use the in-context learning capabilities of GPT-3.5 to summarize these passages and refine the existing knowledge.

Specifically, we use prompt template "*Context: $\{p_{i,j,k}^{rag}\}$; Prompt_input: $\{x_{i,j}^l\}$; Prompt_output:*" to combine the knowledge from Wikipedia passages and LLM-expanded descriptions together, and induce the GPT-3.5 to provide summarization and refinement. Moreover, given the in-context learning ability of recent LLMs, we provide *few-shot demonstrations* to improve the generation quality. In particular, we add two polished demonstrations containing LLM-expanded description and Wikipedia passages as input, as well as a concrete polished output displayed after "*Prompt_output:*". With those polished demonstrations, denoted as $d$, GPT-3.5 tends to perform better, as they help prevent the model from generating irrelevant information. Examples of such manually optimized demonstrations can be found in Appendix D. The final summarized and refined descriptions for each class $c_i$ are denoted as $x_{i,j,k}^d = \text{L}(d, x_{i,j}^l, p_{i,j,k}^{rag})$ where $j \in [N], k \in [N_{rag}]$.

## 3.3 IMAGE GENERATION WITH CONTROLLED DIVERSITY

In this section, we use the final knowledge-enhanced class descriptions $\{x_{i,j,k}^d\}$ as the prompts for text-to-image diffusion model to generate diverse images.

**Image generation with enhanced diversity.** We use diffusion model D, such as Stable Diffusion (Rombach et al., 2022), GLIDE (Nichol & Dhariwal, 2021), and DALLE-3 (Betker et al., 2023), to generate $N_m$ images $m_{i,j,k,q} = \text{D}_q(x_{i,j,k}^d)$ for each prompt $x_{i,j,k}^d$, where $q \in [N_m]$.

To increase image diversity, we alter the parameter "guidance scale" (Rombach et al., 2022) in the diffusion pipeline to control the balance between the precision of the generated image matching the provided prompt and the generation diversity. Since knowledge-enabled prompts already possess a considerable degree of diversity, and too much diversity could lead to noisy generation and hurt performance (Fan et al., 2023), we do not employ additional methods to increase image diversity. In fact, Shipard et al. (2023) suggests additional tricks for improving synthetic diversity, such as generating stylized images. As their initial prompts are not good, they rely on more image generation tricks to improve diversity. However, in our experiments, we found that adding more tricks is not effective. For example, incorporating stylized images doubles the training dataset size, but the accuracy does not significantly improve and rather decreases in some datasets (see Appendix E).

**Selecting high-quality images.** It is unavoidable that some extracted knowledge texts may not be relevant to the targeted class, or some generated images may be of poor quality. Here, we utilize the CLIP score (Radford et al., 2021) to filter out low-quality images. More specifically, for each generated image $m_{i,j,k,q}$, we use the CLIP text embedding $x_i^{temp}$ for the corresponding class name $c_i$, where $x_i^{temp}$ denote the OpenAI suggested prompt templates for CLIP zero-shot classification.[1] Then, we calculate its cosine similarity with the image embeddings as CLIP score. We filter out images with low scores and obtain the filtered images for fine-tuning: $\{m_{i,j,k,q} | \cos(\text{CLIP}(m_{i,j,k,q}), \text{CLIP}(x_i^{temp})) \geq \theta\}$, where CLIP denotes the CLIP encoder for extracting text or image embedding, and $\theta$ is the threshold.

It is worth noting that our primary goal in using CLIP scores is not to *perform precise quality ranking, but rather to eliminate obviously mismatched samples or failed generations for the targeted class*. We observed that this filtering successfully removes two major types of low-quality samples. (1) *Inadequate Text Refinement*: GPT-3.5 occasionally fails to enhance the ConceptNet relations( Section 3.1) due to errors in the knowledge text. This leads to responses like "This sentence is incorrect and does not make sense", resulting in ineffective prompts and unusable synthetic images. (2) *Failed Synthetic Image Generation*: Due to the randomness of diffusion model generation, synthetic images sometimes fail to meet the specific dataset requirements. For example, synthetic images in the EuroSAT dataset did not resemble actual satellite images. We provide examples of such failure cases on ImageNet (Figure 4) and Eurosat (Figure 5) in Appendix F.

---

[1] https://github.com/openai/CLIP/blob/main/data/prompts.md

### 3.4 DOWNSTREAM MODEL FINE-TUNING

We apply the knowledge-enabled synthetic data to improve downstream tasks. Notably, as no original training data is used in our framework, our evaluation belongs to the *zero-shot setting*, demonstrating the versatility applicability of KnowData that only relies on the targeted class names.

**Zero-shot image classification setup.** We focus on improving CLIP models on downstream tasks, given the wide adoption of CLIP for multimodal learning. Considering the potential label space mismatch between CLIP pre-training and the zero-shot downstream task, fine-tuning pretrained CLIP models on our knowledge-enabled dataset can enhance the capabilities.

**Fine-tuning method.** Prior work suggests that finetuning a classifier head based on the frozen pre-trained encoders is sufficient to adapt CLIP to a new task (Wortsman et al., 2022; He et al., 2023). However, in our experiments, we find that we achieve better results by fine-tuning part of the pre-trained image encoder parameters in addition to the classification head. In fact, we believe that knowledge-enhanced data contains more information compared to other baseline synthetic data, and merely fine-tuning the classification head is insufficient for the model to fully learn this content. Therefore, more parameters must be unlocked for the model to learn the distribution. We believe that with the increase in the amount of knowledge-enhanced synthetic data and the richness and accuracy of the knowledge in the data, we will eventually be able to fine-tune the entire pre-trained encoder with better results, which we leave for future work.

## 4 EXPERIMENT

### 4.1 SETUPS

**Datasets.** We use nine datasets, covering object-level, fine-grained, and robustness for zero-shot image classification. *(1) Object-level* includes: (a) Cifar100 (Krizhevsky et al., 2009): extension of the CIFAR-10 dataset to 100 classes, containing low-resolution images. (b) ImageNet (Deng et al., 2009)): a large-scale dataset designed for use in visual object recognition software research, containing high-resolution images (abbreviated as 'IN-Val'). *(2) Fine-grained* includes: (a) DTD (Cimpoi et al., 2014): a collection of textural images in the wild. (b) Eurosat (Helber et al., 2019): a collection of satellite images covering 13 spectral bands and consisting of 10 classes. *(3) Robustness* includes: (a) ImageNet-V2 (Recht et al., 2019): a reproduction of the ImageNet with distribution shift (abbreviated as 'IN-V2'). (b) ImageNet-Sketch (Wang et al., 2019): black and white sketches of ImageNet (abbreviated as 'IN-Sketch'). (c) ImageNet-R (Hendrycks et al., 2021a): renditions (e.g.,art, patterns, etc.) of 200 ImageNet classes (abbreviated as 'IN-R'). (d) ObjectNet (Barbu et al., 2019): real-world objects from ImageNet with diversity. (e) ImageNet-A (Hendrycks et al., 2021b): ImageNet with naturally occurring examples filtered (abbreviated as 'IN-A').

**Models.** We use GPT-3.5 (Brown et al., 2020) for generating and summarizing knowledge descriptions. By default, we employ the Stable Diffusion (`stable-diffusion-v1-5` endpoint) (Rombach et al., 2022) for image generation, and we additionally evaluate GLIDE (Nichol & Dhariwal, 2021) and DALLE-3 (Azure OpenAI API) (Betker et al., 2023) in ablation studies. For fine-tuning on the synthetic data for zero-shot classification, we use two pre-trained CLIP models: CLIP-RN50 based on ResNet-50 (He et al., 2016) and CLIP-ViT-B/16 based on ViT-B/16 (Dosovitskiy et al., 2020). We fine-tune these models using cross-entropy loss, with a learning rate of 1e-5, a weight decay of 0.1, and for 15 epochs. Specifically, we fine-tune the last 31 layers for CLIP-ViT-B/16 and the last 44 layers for CLIP-RN50 (details on selecting the layers to fine-tune are provided in Appendix G.)

**Synthetic dataset details.** We generate $480k$ synthetic images based on ImageNet class names to fine-tune the downstream models, and then evaluate the fine-tuned models on the ImageNet test data and its out-of-distribution variants. We generate about $60k$ images for other datasets with fewer categories, including CIFAR100, DTD, and EuroSAT. The detailed number of synthetic prompts and images corresponding to each stage in our pipeline for different datasets can be found in Table 1. We use 10 NVIDIA RTX A6000 to perform data generation. Generating $60k$ data requires 12 hours.

**Baselines.** We consider the OpenAI's pretrained CLIP models and 5 state-of-the-art CLIP zero-shot methods in the two categories discussed in Section 2 as our baselines. Specifically, *(1) among baselines that enhance the initial text embeddings*, we evaluate: (a) **ZPE** (Allingham et al., 2023), which establishes a pool of templates and then improves zero-shot results by using weighted selection

Table 1: Synthetic dataset size at different stages in KnowData.

| Dataset | # class | # prompts | | | | # images | |
|---|---|---|---|---|---|---|---|
| | | after ConceptNet | after GPT expansion | after Wiki RAG | after GPT summarization | diffusion model generated | after CLIP score filtering |
| CIFAR100 | 100 | $100 \times 100 = 10000$ | 10000 | $2 \times 10000 = 20000$ | 20000 | $4 \times 20000 = 80000$ | $0.75 \times 80000 = 60000$ |
| DTD | 47 | $100 \times 47 = 4700$ | 4700 | $2 \times 4700 = 9400$ | 9400 | $8 \times 9400 = 75200$ | $0.8 \times 75200 = 60160$ |
| EuroSAT | 10 | $100 \times 10 = 1000$ | 1000 | $3 \times 1000 = 3000$ | 3000 | $25 \times 3000 = 75000$ | $0.8 \times 75000 = 60000$ |
| ImageNet&Variant | 1000 | $100 \times 1000 = 100000$ | 100000 | $2 \times 100000 = 200000$ | 200000 | $4 \times 200000 = 800000$ | $0.6 \times 800000 = 480000$ |

Table 2: Zero-shot image classification results based on KnowData compared with SOTA methods. The column **S** indicates the use of synthetic data, **P** denotes the use of pre-trained models, **E** represents the incorporation of external knowledge, and **IN-Avg** is the average accuracy across ImageNet and its variants. **\*** denotes our reproduced results for baselines, and **-** means that the baseline method does not support the evaluation setting. The highest accuracy across all methods is **bolded**[3].

| Model | Method | S | P | E | CIFAR100 | DTD | EuroSAT | IN-Val | IN-V2 | IN-R | IN-A | IN-Sketch | IN-Avg |
|---|---|---|---|---|---|---|---|---|---|---|---|---|---|
| CLIP ViT-B/16 | OpenAI (Radford et al., 2021) | × | √ | × | 68.70 | 46.00 | 54.10 | 68.60 | 61.60* | 77.57* | 50.23* | 48.23* | 61.25 |
| | ZPE (Allingham et al., 2023) | × | √ | √ | 66.63 | 46.28 | 53.82 | 68.60 | 62.21 | 77.62 | 49.63 | 47.99 | 61.21 |
| | Description (Menon & Vondrick, 2023) | × | √ | √ | – | 45.59 | 48.82 | 68.03 | 61.54 | 75.00* | 49.17* | 47.08* | 60.16 |
| | Hierarchy (Ge et al., 2023) | × | √ | √ | 35.20 | – | – | 68.86 | 62.00 | 60.62 | 31.07 | 48.26 | 54.16 |
| | Synthetic (He et al., 2023) | √ | √ | √ | 70.71 | 44.92 | 59.86 | 69.16 | 61.28* | 76.41 | 48.25* | 48.47 | 60.71 |
| | Diversity (Shipard et al., 2023) | √ | × | × | 32.38 | – | 21.71 | – | – | – | – | – | – |
| | KnowData (ours) | √ | √ | √ | **73.88** | **57.51** | **63.86** | **70.44** | **64.13** | **78.20** | 48.65 | **50.63** | **62.41** |
| CLIP RN50 | OpenAI (Radford et al., 2021) | × | √ | × | 41.60 | 41.70 | 41.10 | 59.60 | 52.92* | 60.53* | 22.80* | 35.38* | 46.25 |
| | Description (Menon & Vondrick, 2023) | × | √ | √ | – | 41.90* | 37.58* | 59.59* | 53.02* | 57.20* | **23.55*** | 33.73* | 45.42 |
| | Hierarchy (Ge et al., 2023) | × | √ | √ | – | – | – | 59.76* | 53.11* | 42.59* | 11.21* | 35.55* | 40.44 |
| | Synthetic (He et al., 2023) | √ | √ | √ | 45.69 | 43.19 | 55.37 | 60.78 | 51.14* | 59.37 | 21.91* | 36.55 | 45.95 |
| | Diversity (Shipard et al., 2023) | √ | × | × | 45.63 | – | 39.92 | – | – | – | – | – | – |
| | KnowData (ours) | √ | √ | √ | **57.16** | **51.18** | **57.19** | **61.73** | **54.67** | **60.67** | 19.75 | **37.74** | **46.91** |

among these templates to serve as the classification head. (b) **Description** (Menon & Vondrick, 2023), which uses the description of the label instead of the label name itself as the input for text embedding for classification. (c) **Hierarchy** (Ge et al., 2023), which enhances labels through the WordNet hierarchy for data with low confidence. *(2) Among baselines that involve fine-tuning with synthetic images*, we evaluate: (a) **Synthetic** (He et al., 2023), which enhances labels with the T5 model, generates images using these enhanced labels with the GLIDE model and then fine-tunes only the classification head of CLIP. (b) **Diversity** (Shipard et al., 2023), which utilizes images generated with three different tricks to enhance diversity and fine-tunes a model with random initialization.

**Evaluation metrics.** We use three common metrics to evaluate the quality of our generated images. *(1) Accuracy.* For a test image, we input it into the image encoder to get the image embedding. By multiplying this with the classification head and taking the argmax, we can predict the label for the image. The top-1 accuracy across all images is used to determine the final accuracy. *(2) CLIP score.* Unlike the CLIP score used for filtering in Section 3.3, the text embeddings for the CLIP score here are obtained from our knowledge-enabled prompts, rather than being derived from class names combined with each dataset's CLIP template. This metric reflects the reliability of the generated image regarding the prompt. *(3) Diversity score.* Following Boutin et al. (2023), we compute the standard deviation in the feature space (SimCLR image encoder (Chen et al., 2020)) for images from every class, and then compute the average score across all classes as the diversity score. Specifically, for a given category $j$, composed of $M$ samples and a feature space $f$, the diversity $\sigma_j$ is computed as follows: $\sigma_j = \sqrt{\frac{1}{M-1} \sum_{i=1}^{M}(f(v_i^j) - \frac{1}{M} \sum_{i=1}^{M} f(v_i^j))^2}$, where $v_i^j$ is $i$-the image of class $j$.

## 4.2 EXPERIMENTAL RESULTS

**KnowData improves CLIP's zero-shot performance.** We evaluate KnowData on zero-shot image classification tasks by fine-tuning CLIP ViT-B/16 and CLIP RN50 models on our generated synthetic data. (1) The results in Table 2 show that, on ViT-B, compared to the best SOTA methods, KnowData achieves 11.23% and 4% performance improvements on the DTD and EuroSAT datasets, respectively. On RN50, the performance improvements were 11.47% and 7.99% on the Cifar100 and DTD datasets, respectively. (2) Besides significant improvements on fine-grained datasets, our results on the ImageNet and its variants consistently surpassed those of SOTA methods. For example, on In-Val dataset with CLIP ViT-B/16 model, the previous SOTA method, Synthetic (He et al., 2023), only achieves 0.56% accuracy improvement over OpenAI CLIP baseline, whereas KnowData reached 1.28% accuracy improvement. This demonstrates the effectiveness of our knowledge-enabled data generation pipeline. Moreover, other SOTA methods performed poorly on individual ImageNet variant datasets, failing to exceed the overall performance of OpenAI CLIP, while our model, fine-tuned with the *same* set of synthetic data, showed better performance cross various

ImageNet variant datasets, proving the out-of-distribution robustness of knowledge empowerment in enhancing zero-shot capabilities.

It is noteworthy that, we have reproduced and compared results from various SOTA CLIP zero-shot classification methods, unlike existing works (Allingham et al., 2023; Menon & Vondrick, 2023; Ge et al., 2023; He et al., 2023) that only compare to OpenAI CLIP baseline. Our results set stronger baselines for evaluation and enable a more comprehensive understanding of related research.

**KnowData produces synthetic images with better reliability and diversity.** In addition to the accuracy evaluated above, we further evaluate the CLIP score and diversity score of synthetic images. In particular, we focus on how different components of KnowData that aim to gradually improve the text descriptions, affect the image quality metrics compared to the base prompt ("BP"). In Table 3,

(1) the CLIP score reflects the alignment between the image and text, ensuring that the image accurately represents the content intended by the text. The results in Table 3 show that as knowledge gradually enriches, the CLIP score tends to increase, indicating that knowledge can improve the reliability of synthetic data, enabling it to generate the content intended by the text more accurately. We note that the decrease when we add Wiki RAG knowledge ("+WRAG") is mainly due to the CLIP text encoder's inherent limitation of handling inputs within 77 tokens, leading to the truncation of lengthy texts. (2) Furthermore, the diversity score, calculated using the standard deviation in the feature space for images, also increases as knowledge enriches, demonstrating that the addition of knowledge can also serve as a diversity trick, allowing for generating more rich and varied images.

Table 3: The components in KnowData improve the CLIP score and the diversity of synthetic data. **BP**: the baseline using base prompt "*A photo of $\{c_i\}$*". The components of KnowData include: **CN**, adding ConceptNet knowledge; **GPT**, adding GPT expansion; **WRAG**, adding RAG based on Wikipedia.

| Method | CLIP Score | Diversity |
|---|---|---|
| BP | 0.3274 | 31.13 |
| CN | 0.3409 | 31.93 |
| CN+GPT | **0.3641** | 33.73 |
| CN+WRAG+GPT | 0.3513 | **37.02** |

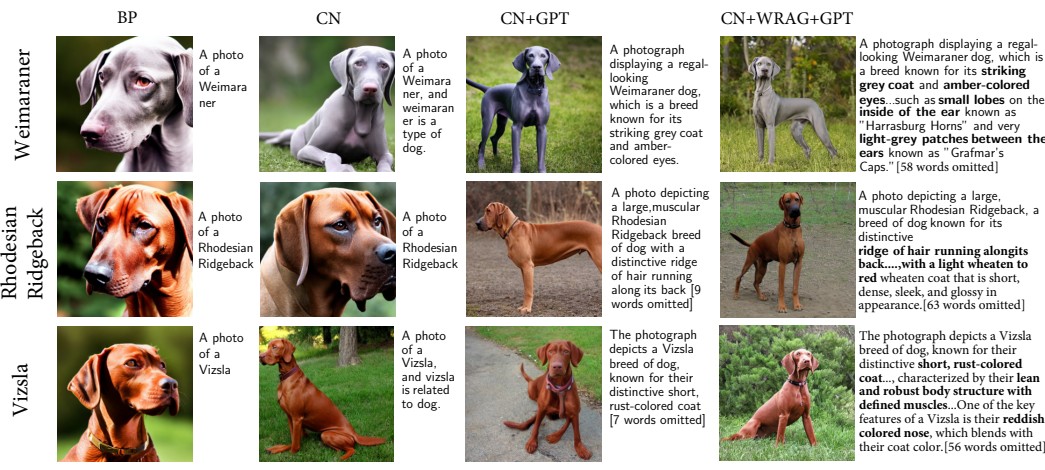

Figure 2: Stable Diffusion generated images for three similar types of dogs (Weimaraner, Rhodesian Ridgeback, Vizsla) given different prompts. KnowData incorporating the knowledge from Concept-Net (**CN**), GPT-3.5 (**GPT**) and Wikipedia (**WRAG**) can generate images of complete objects with better details. However, objects from different classes are less distinguishable under base prompt (**BP**).

**KnowData produces synthetic images with more accurate details and more diverse background.** In Figure 2 and Appendix Figures 6 to 8, we display pairs of image-text from different numbers of knowledge sources and annotate helpful information in the prompt from each knowledge source. In summary, we find that knowledge-enabled image generation can *1) provide a more complete view*

---

[3]We have filled in the table as much as possible, and reproduced the results for datasets that are not included in the original paper of previous methods to our best. Some dataset/method combinations are difficult to reproduce due to the absence of crucial knowledge or prompts. We mark such cells with "-".

Table 4: KnowData achieves better results with stronger data generators.

| Data generator | IN-Val | IN-V2 | IN-R | IN-A | IN-Sketch | ObjectNet | Average |
|---|---|---|---|---|---|---|---|
| GLIDE | 67.64 | 61.19 | 76.29 | 47.25 | 48.34 | 51.79 | 58.75 |
| Stable Diffusion | **69.85** | **63.48** | 78.16 | **49.24** | 49.87 | **55.10** | 60.95 |
| DALLE-3 | 69.66 | 62.93 | **78.81** | 48.48 | **51.47** | 54.76 | **61.02** |

Table 5: Ablation study on diversity in knowledge sources and image generation. **BP**: using base prompt "*A photo of {$c_i$}*", **CN**: adding ConceptNet knowledge, **GPT**: adding GPT expansion, **Div**: using image diversity tricks, **WRAG**: using RAG based on external knowledge store Wikipedia.

| Model | Method | DTD | EuroSAT | IN-Val | IN-V2 | IN-R | IN-A | IN-Sketch | ObjectNet | Average |
|---|---|---|---|---|---|---|---|---|---|---|
| | BP | 48.11 | 54.19 | 69.04 | 62.57 | 77.85 | 48.44 | 49.61 | 54.68 | 60.37 |
| | BP+Div | 49.17 | 57.90 | 69.64 | 63.03 | 77.88 | 48.08 | 49.80 | 54.93 | 60.56 |
| CLIP ViT-B/16 | CN+Div | 53.01 | 60.94 | 69.48 | 62.97 | 77.92 | 48.75 | 49.75 | 54.55 | 60.57 |
| | PureGPT+Div | 53.84 | 55.91 | 69.89 | 63.31 | 77.93 | 48.40 | 49.93 | 54.47 | 60.66 |
| | CN+GPT+Div | 55.85 | 62.30 | 69.63 | 63.08 | 78.07 | 48.68 | 49.82 | 54.86 | 60.69 |
| | CN+WRAG+GPT+Div | **57.33** | **63.86** | **69.95** | **63.61** | **78.18** | **48.81** | **49.93** | **55.36** | **60.97** |

*of the object, 2) present more accurate details to help differentiate similar classes, and 3) produce more diverse backgrounds*. Take Figure 2 with three similar dog species (Weimaraner, Rhodesian Ridgeback, and Vizsla) as an example. We see that with the base prompt, the generated images can distinguish Weimaraner but cannot differentiate between Rhodesian Ridgeback and Vizsla. However, with the addition of knowledge, the generated images can differentiate them through the distinct coat colors (Rhodesian Ridgeback with light wheaten or red wheaten coat, and Vizsla with rust-colored coat) and the unique nose color of Vizsla (reddish-colored nose, which blends with their coat color). Moreover, it is evident that the images generated by the base prompt have a very uniform style of dogs (e.g., showing only the head), while with the addition of knowledge, their poses and backgrounds become increasingly rich and the full body of the dogs are displayed, making the images more realistic.

**KnowData benefits from stronger data generators.** In our experiments, we used open-source Stable Diffusion for synthetic image generation. Here, we study the effect of data generators and additionally evaluate DALLE-3 and GLIDE. We fine-tune CLIP-ViTB/16 with $60k$ images generated by different text-to-image generators using KnowData and evaluate the accuracy on ImageNet-Val and its 5 variant testsets. The results in Table 4 show that stronger data generators (Stable Diffusion and DALLE-3 compared to GLIDE) improve zero-shot performance through knowledge-enabled data. It demonstrates the potential of KnowData as the community builds stronger data generators. While both DALLE-3 and Stable Diffusion offer strong performance, we primarily use the open-source model Stable Diffusion in our experiments due to convenience and efficiency.

**KnowData utilizes data more efficiently when scaling synthetic data size.** To study the data scaling law, from the synthetic data filtered by CLIP score (Section 3.3), we randomly sample $10\% \sim 100\%$ (in $10\%$ increments) of the data to fine-tune the downstream model. The results on averaged accuracy on ImageNet and its variants (left) and accuracy on DTD dataset (right) in Figure 3 show that KnowData not only surpasses the base prompt method but also shows more noticeable improvement as the volume of data increases, demonstrating better data scaling ability.

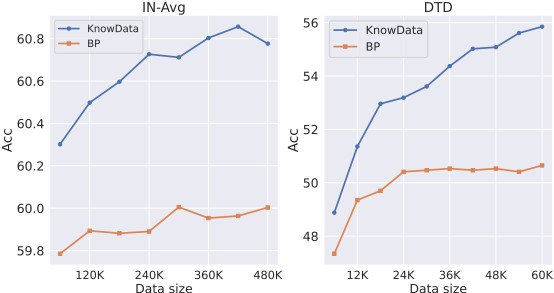

Figure 3: KnowData demonstrates better data scaling law than the base prompt (BP) method in terms of average accuracy on ImageNet (In-Val and 5 variants) and accuracy on DTD.

**Diversity in knowledge and diversity in images both matter.** In Table 5, we conduct ablation studies on the components of KnowData. Here we fine-tune CLIP-ViTB/16 with $80k$ synthetic ImageNet images, and $60k$ synthetic images for other datasets. (1) From using base prompts (BP), to adding ConceptNet knowledge (+CN), then to incorporating GPT extensions and summaries (+GPT), and finally to adding Wikipedia-based retrieval augmented generation (+WRAG), we see continuous accuracy improvement, underscoring the importance of diverse knowledge sources and text quality in

KnowData. (2) Additionally, the diversity of images is also crucial, as evidenced by the comparison between using and not using diversity techniques (+Div) in the first and second rows. (3) Furthermore, we consider the Pure GPT baseline where we directly prompt GPT-3.5 to generate descriptions about classes (using prompts "*write a detailed description about {$c_i$}*"). The results show that the Pure GPT baseline performs worse than KnowData that incorporates external knowledge sources, including ConceptNet and Wikipedia. It indicates that the descriptions generated by the GPT-3.5 could lack authenticity and diversity due to the potential LLM hallucinations. Explicitly injecting structured knowledge as in KnowData can help improve both accuracy and diversity.

**KnowData can benefit downstream models with difference sizes.** In addition to CLIP ViT-B/16, we conduct experiments on the larger downstream models such as CLIP ViT-L/14 pretrained on WebImageText

Table 6: Evaluation on different sizes of downstream models.

| Dataset | Method | ViT-B/16 | ViT-L/14 | ViT-G/14 |
|---------|--------|----------|----------|----------|
| CIFAR100 | OpenAI (Radford et al., 2021) | 68.70 | 78.30 | 83.97 |
| | ZPE (Allingham et al., 2023) | 66.63 | 79.36 | - |
| | KnowData (ours) | **73.88** | **83.42** | **85.70** |

(WIT) (Radford et al., 2021) and ViT-G/14 pretrained on LAION-2B (Schuhmann et al., 2022) The results in Table 6 show that the model fine-tuned on KnowData generated synthetic data performs better than pre-trained CLIP (+5.12%) and the SOTA method ZPE (Allingham et al., 2023) (+4.06%) on ViT-L/14, and also surpasses pretrained CLIP (+1.66%) on ViT-G/14[4] It suggests that even models pre-trained on large datasets with relatively high zero-shot accuracy can still benefit from KnowData's fine-tuning by a noticeable margin. This indicates that large pre-training datasets might still lack relevant knowledge (e.g., images of certain knowledge might be rare on the internet and thus insufficient in pre-training). KnowData retrieves a comprehensive set of knowledge from ConceptNet, Wikipedia and LLM, and generates corresponding images to supplement the knowledge.

**Fine-tuning CLIP on KnowData improves downstream task performance on VQA and WinoGround.** In addition to image-classification task, KnowData can potentially benefit other downstream tasks. We follow the existing evaluation method (Shen et al., 2021) to evaluate zero-shot per-

Table 7: Zero-shot performance of CLIP models on VQA v2 (Goyal et al., 2017) and WinoGround (Thrush et al., 2022) downstream tasks.

| Model | VQA v2 | WinoGround | | |
|-------|--------|------------|---|---|
| | Accuracy | Text score | Image score | Group score |
| Pretrained CLIP (Radford et al., 2021) | 51.62 | 25.25 | 10.25 | 7.00 |
| KnowData-finetuned CLIP | **54.86** | **27.50** | **12.00** | **8.00** |

formance of CLIP model on VQA v2 dataset (Goyal et al., 2017). Specifically, we append each label used in VQA v2 to the corresponding question in the format *[Question]+[Label]* and then calculate zero-shot performance by matching the most similar label to the question's *[image]* embedding. Following (Shen et al., 2021), we evaluate the "yes/no" questions (with question type "Are these...") on *VQA v2 mini-eval*. We compare the performance of pretrained ViT-B/16 CLIP against the CLIP model finetuned using KnowData synthetic data generated from ImageNet class labels. As shown in Table 7, KnowData-finetuned model can be more generalized with improved accuracy on VQA v2 task.

Besides, we evaluate KnowData on WinoGround benchmark (Thrush et al., 2022) which require explicit composition abilities. As shown in Table 7, KnowData fine-tuned CLIP model improves text, image, and group scores. It indicates that the fine-tuned encoders have enhanced composition abilities, allowing them to better discern similar image-text pairs in WinoGround. Given the distinct nature of the knowledge from ImageNet class labels and WinoGround tasks, the results reflect improved generalization capabilities of the fine-tuned model.

## 5 CONCLUSION

In this work, we propose a knowledge-enabled image-text pairs generation framework, KnowData, which leverages real-world knowledge from ConceptNet and Wikipedia, along with large language models and advanced text-to-image models. Our extensive evaluation results show that our approach leads to better CLIP zero-shot performance across various domains, highlighting the importance of integrating diverse knowledge sources for enhancing multimodal learning models.

---

[4]The result of ZPE on ViT-G/14 is not available in its paper, and ZPE's implementation is not open-sourced.

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

## A  Broader Impact

**Positive Societal Impacts**  The framework presented in our paper, KnowData, offers several positive societal impacts, particularly in advancing the capabilities of multimodal models such as CLIP: (1) *Enhanced Learning and Accessibility*: By integrating real-world knowledge from sources like knowledge graphs and Wikipedia, KnowData produces more contextually rich and accurate text descriptions. This can improve the educational value and accessibility of AI-generated content, making it more informative and beneficial for users. (2) *Improved CLIP Performance*: Our approach enhances the performance of CLIP models in zero-shot image classification tasks, as demonstrated by significant performance improvements across multiple datasets. This can lead to more robust and adaptable CLIP-based systems that perform better in real-world applications. (3) *Promotion of Multimodal Research*: The successful integration of structured and unstructured knowledge into text descriptions can inspire further research in the integration of diverse data sources for multimodal learning, fostering innovation and progress in the field.

**Negative Societal Impacts and Mitigation Strategies**  KnowData integrate real-world knowledge into text descriptions, thereby enhancing CLIP model performance compared to using purely LLM-generated text descriptions. While knowledge sources like Wikipedia and ConceptNet are widely acknowledged as reliable text sources, our framework still relies on LLMs to summarize and refine the text descriptions. This reliance introduces the possibility that the LLMs may inadvertently introduce biases or fairness issues affecting certain groups. To mitigate this, future work is needed to research and apply techniques to detect and reduce biases in the generated content.

## B  Limitations and Future Work

In our work, we use knowledge-enabled description to generate synthetic images, and use synthetic images to fine-tune the CLIP models for zero-shot image classification tasks.

(1) As discussed in Section 3.4, we found that fine-tuning partial layers of the image encoder performs better than fine-tuning the entire image encoder. Achieving higher quality image generation, which might enable effective fine-tuning of the entire image encoder, still requires more comprehensive and accurate knowledge integration in the future.

(2) The fine-tuning process incurs additional computation costs compared to using pretrained CLIP models. If efficiency is a constraint, the knowledge-enabled texts generated by our KnowData can be used to directly enhance per-class text embeddings for CLIP image classification without fine-tuning. We leave the exploration of this approach for future work.

(3) Another future work would be comparing our generated text captions with normally collected ones (e.g., web-crawled image captions). We note that crawling high-quality web image captions and selecting the most relevant ones for each class label is a challenging and non-trivial task, which could itself constitute a novel contribution and is a promising future research direction.

(4) While our work primarily focuses on evaluating generated synthetic data on downstream image classification/VQA tasks, extending our evaluation to improve other vision-language capabilities of CLIP, including text/image retrieval, is an important and exciting direction.

## C  ConceptNet Knowledge

We focus on 18 relations from ConceptNet:   "RelatedTo","FormOf","IsA","PartOf", "HasA","UsedFor","CapableOf",  "AtLocation","HasProperty","CreatedBy","SymbolOf",  "DefinedAs","LocatedNear","HasContext","SimilarTo", "MadeOf", "CausesDesire","ReceivesAction".

To convert these relations into more understandable sentences, we use the templates: "{} is related to {}","{} is a form of {}","{} is a type of {}","{} is a part of {}","{} has {} {}","{} is used for {}","{} is capable of {}","{} is at the location of {}","{} can be described as {}","{} is created by {}","{} symbolically represents {}",'{} and {} overlap considerably in meaning, and {} is a more explanatory version of {}","{} and {} are typically found near each other","{} is a word used in the context of {}","{} is similar to {}","{} is made of {}","{} makes someone want {}","{} can be done to {}".

## D   IN-CONTEXT LEARNING METHOD FOR RETRIEVAL AUGMENTED GENERATION

We use the following template to guide GPT in summarizing the content of passages retrieved and to adjust and supplement the original prompt.

```
{example0}
{example1}
--------------------
Context:
{passage_input}
Prompt input:
{prompt_input}
Prompt output:
```

The examples were manually polished by us, totaling 20 in number. For each prompt, we randomly select 2 to be incorporated into the aforementioned template, which, along with the sentence itself and the passage retrieved, guide GPT in the generation process. Here, we showcase two of these examples.

```
--------------------
Context:
Tincinae Tincinae is a subfamily of freshwater ray-finned fish from the family Cyprinidae, it consists of the tench
of Eurasia and the east Asian clod minnows. Tinca tinca is a freshwater tincinae fish that is found in the Danube
basin
Prompt input:
A photo of a tench, which is a freshwater fish belonging to the family cyprinidae characterized by its olive-green
coloration and small scales.
Prompt output:
A photo of a Tinca tinca, a freshwater tench from the Tincinae subfamily within the Cyprinidae family, characterized
by its olive-green coloration and small scales, native to the Danube basin in Eurasia.
--------------------
Context:
Goldfish The Goldfish (Carassius auratus) is a freshwater fish in the family Cyprinidae of order Cypriniformes.
Goldfish breeds vary greatly in size, body shape, fin configuration, and coloration (various combinations of white,
yellow, orange, red, brown, and black are known). Native to China, the goldfish is a relatively small member of the
carp family (which also includes the Prussian carp and the crucian carp). It is commonly kept as a pet in indoor
aquariums, and is one of the most popular aquarium fish. Goldfish released into the wild have become an invasive
pest in parts of North America. It was first selectively bred for color in imperial China more than 1,000 years ago,
where several distinct breeds were developed.
Prompt input:
A photograph capturing the image of a small, bright orange goldfish, a freshwater fish species belonging to the
family Cyprinidae known for their distinctive scales and long fins.
Prompt output:
A photograph capturing a small, bright orange Carassius auratus, commonly known as a goldfish, a popular freshwater
species from the Cyprinidae family, renowned for its distinctive scales and long fins, and a history of over 1,000
years of selective breeding for varied colorations in China.
```

The reason for this approach is that if we use designed prompts to guide GPT in generation, it can lead to the production of some irrelevant information, as in the example below: "*This sentence accurately describes a goldfish.*"

```
Goldfish The Goldfish (Carassius auratus) is a freshwater fish in the family Cyprinidae of order Cypriniformes.
Goldfish breeds vary greatly in size, body shape, fin configuration, and coloration (various combinations of white,
yellow, orange, red, brown, and black are known). Native to China, the goldfish is a relatively small member of the
carp family (which also includes the Prussian carp and the crucian carp). It is commonly kept as a pet in indoor
aquariums, and is one of the most popular aquarium fish. Goldfish released into the wild have become an invasive
pest in parts of North America. It was first selectively bred for color in imperial China more than 1,000 years ago,
where several distinct breeds were developed.
--------------------
Given the context information above, please extract the most related information for the following sentence:
"A photograph capturing the image of a small, bright orange goldfish, a freshwater fish species belonging to the
family Cyprinidae known for their distinctive scales and long fins.";
and please append the extracted information to the end of this sentence to provide more details for this sentence
while maintaining its semantic meaning.

This sentence accurately describes a goldfish, a freshwater fish in the family Cyprinidae known for their distinctive
scales and long fins. Goldfish breeds vary greatly in coloration, and some may be bright orange in color. The
goldfish is commonly kept as a pet in indoor aquariums and was first selectively bred for color in imperial China
over 1,000 years ago. However, goldfish released into the wild have become an invasive pest in parts of North America.
```

## E   CONTROLLING THE DEGREE OF IMAGE DIVERSITY

Given the final prompts from KnowData empowered with knowledge through ConceptNet, Wikipedia, and GPT-3.5, we evaluate two image generation methods: (1) **Random guidance scale**: it is achieved by randomly setting the "guidance scale" parameters in the Stable Diffusion pipeline to control the balance between the precision of the generated image matching the provided prompt and the generation diversity. (2) **Stylization** and random guidance scale: we further add stylization to

diversify the prompts, which is achieved by adding corresponding style descriptions (such as photo, painting, artwork, sketch, collage, poster, digital art image, rock painting, doodle, 3D rendering) to the prompt templates, so as to enhance the diversity of image generation. The number of images generated for these two methods are 0.6M and 1.2M, respectively.

The results in Table 8 show that the accuracy does not increase but decreases despite the doubling in the size of the training dataset. To facilitate subsequent testing and achieve better results, we choose to use the randomized guidance scale as our method to enhance image diversity.

Table 8: Adding stylization during image generation process does not necessarily improve the overall synthetic image quality for ImageNet, as reflected by the zero-shot performance of fine-tuned downstream CLIP RN50 models.

| Model | Method | # Synthetic images | IN-Val | IN-V2 | IN-R | IN-A | IN-Sketch | ObjectNet |
|---|---|---|---|---|---|---|---|---|
| CLIP RN50 | Random guidance scale | 0.6M | **61.33** | **54.17** | 60.61 | **23.04** | 35.89 | **47.01** |
| | Stylization + Random guidance scale | 1.2M | 61.10 | 53.89 | **60.63** | 22.75 | **36.00** | 46.87 |

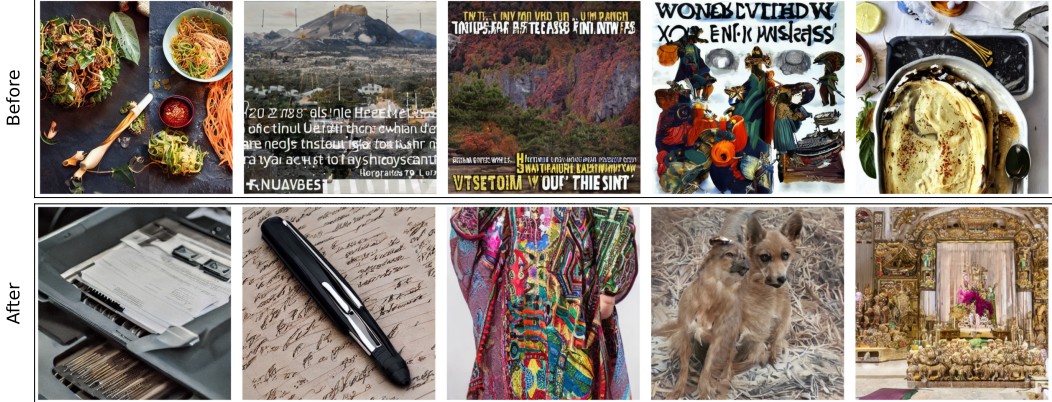

Figure 4: Five images with the lowest CLIP scores from synthetic ImageNet dataset before and after applying CLIP score filtering.

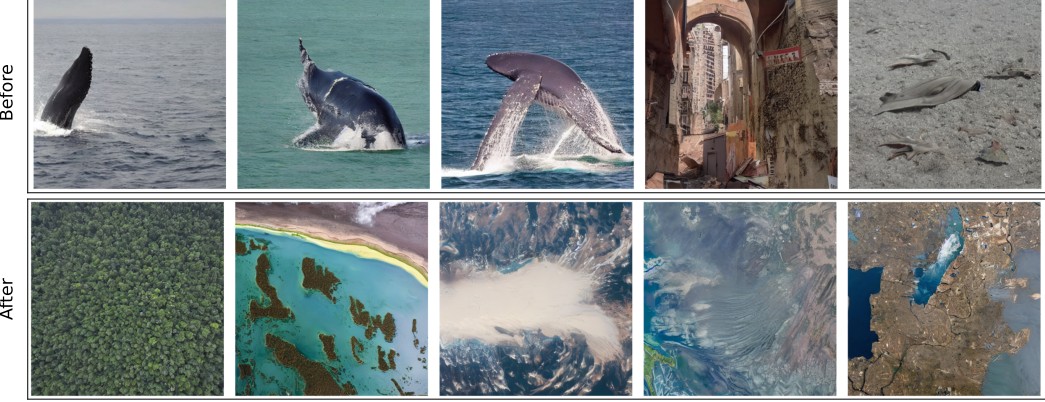

Figure 5: Five images with the lowest CLIP scores from synthetic EuroSAT dataset before and after applying CLIP score filtering.

# F   CLIP Score Filtering Results

We present the five images with the lowest CLIP scores from ImageNet (see Figure 4) and EuroSAT (see Figure 5) before and after applying CLIP score filtering. We identify two major failure patterns in low-quality images before filtering: (1) *Inadequate Text Refinement*: GPT-3.5 occasionally fails to enhance the ConceptNet relations( Section 3.1) due to errors in the knowledge text. This leads to responses like "This sentence is incorrect and does not make sense", resulting in ineffective prompts and unusable synthetic images. (2) *Failed Synthetic Image Generation*: Due to the randomness of diffusion model generation, synthetic images sometimes fail to meet the specific dataset requirements. For example, synthetic images in the EuroSAT dataset did not resemble actual satellite images.

After applying CLIP score filtering, images that did not meet the overall dataset requirements (such as the necessary satellite images in EuroSAT) and those generated from inappropriate text descriptions (as in ImageNet) were effectively filtered out. This process significantly improved the quality and relevance of the remaining images.

# G   Details on CLIP Model Fine-tuning

We treat the number of model layers in the pretrained CLIP model to fine-tune as a hyperparameter. Starting from the classification head, we gradually unfreeze more blocks in the image encoders to fine-tune while keeping the remaining layers frozen. As shown in Table 9, the ViT-B/16 model performs best when fine-tuning the last 31 layers (including the classification head) with a $480k$ ImageNet synthetic dataset. Therefore, we choose to fine-tune the last 31 layers for evaluation on the CIFAR, EuroSAT, and ImageNet variant datasets. Similarly, for the RN50 model, we choose to fine-tune the last 44 layers.

It is noteworthy that although we select the layers to fine-tune based on the results from the ImageNet validation dataset, the evaluation results on other ImageNet variant datasets show that the chosen layers consistently yield better performance across multiple datasets.

Table 9: Performance on IN-Val, IN-V2, IN-R, IN-A, IN-Sketch and ObjectNet when fine-tuning different numbers of layers in pretrained CLIP ViT-B/16 on KnowData generated synthetic ImageNet data.

| Model | Number of fine-tuning layers | Layers description | IN-Val | IN-V2 | IN-R | IN-A | IN-Sketch | ObjectNet |
|---|---|---|---|---|---|---|---|---|
| CLIP ViT-B/16 | last 2 layers | Classificaiton Head | 69.64 | 63.01 | 77.83 | **50.81** | 49.11 | 54.71 |
| | last 7 layers | LayerNorm+Classificaiton Head | 68.98 | 62.15 | 77.44 | 49.51 | 48.22 | 54.20 |
| | last 19 layers | 11th Block+LayerNorm+Classificaiton Head | 70.22 | 63.39 | 77.87 | 49.04 | 49.44 | 54.93 |
| | last 31 layers | 10-11th Blocks+LayerNorm+Classificaiton Head | **70.41** | **63.95** | 78.25 | 48.84 | 49.70 | **55.00** |
| | last 43 layers | 9-11th Blocks+LayerNorm+Classificaiton Head | 70.34 | 63.40 | **78.49** | 48.41 | **49.98** | 54.87 |
| | last 55 layers | 8-11th Blocks+LayerNorm+Classificaiton Head | 70.15 | 63.04 | 77.58 | 48.00 | 49.70 | 54.22 |

Table 10: Performance on IN-Val, IN-V2, IN-R, IN-A, and IN-Sketch when fine-tuning different numbers of layers in pretrained CLIP RN50 on KnowData generated synthetic ImageNet data.

| Model | Number of fine-tuning layers | Layers description | IN-Val | IN-V2 | IN-R | IN-A | IN-Sketch |
|---|---|---|---|---|---|---|---|
| CLIP RN50 | last 2 layers | Classificaiton Head | 60.24 | 53.16 | 60.35 | **22.39** | 35.35 |
| | last 44 layers | 4th Block+AttentionPool+Classificaiton Head | **61.75** | 54.44 | **60.48** | 20.19 | **36.85** |
| | last 101 layers | 3-4th Blocks+AttentionPool+Classificaiton Head | 61.73 | 54.37 | 59.89 | 18.15 | 36.20 |
| | last 140 layers | 2-4th Blocks+AttentionPool+Classificaiton Head | 61.55 | **54.57** | 59.81 | 17.80 | 36.41 |
| | last 170 layers | 1-4th Blocks+AttentionPool+Classificaiton Head | 61.58 | 54.42 | 59.80 | 17.88 | 36.34 |

# H   Examples of Knowledge-Enabled Generation of Text-Image Pairs

In Figures 6 to 8, we display pairs of images and their corresponding text generated from varying numbers of knowledge sources. Each prompt is annotated to highlight the helpful information contributed by each knowledge source. We find that knowledge-enabled image generation can: 1)

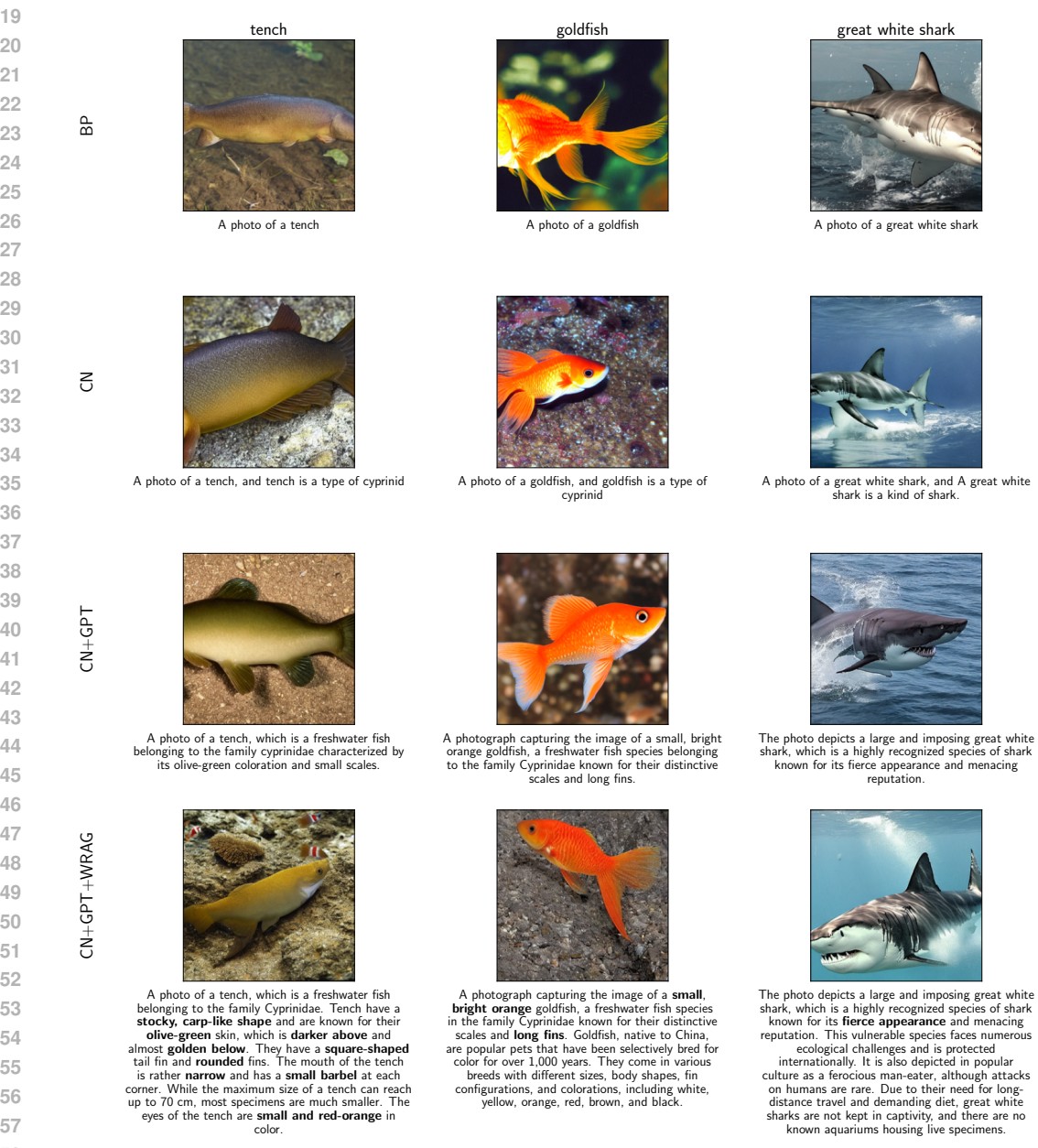

Figure 6: Examples of generated image-text pairs for fish.

Provide a more complete view of the object. 2) Present more accurate details to help differentiate similar classes. 3) Produce more diverse backgrounds.

In the example of fish (Figure 6), it can be seen that fish generated using KnowData can display their complete form, while images generated from the base prompt only show the tail. In the example of birds (Figure 7), it is evident that the background of the KnowData generated images becomes more enriched as knowledge increases. In the example of non-animal objects (Figure 8), additional details can also be seen in KnowData that distinguish between similar types, such as the Acoustic guitar and the Electric guitar. These two types of objects initially have similar backgrounds, but later, the Electric guitar includes an amplifier when Wikipedia knowledge is incorporated.

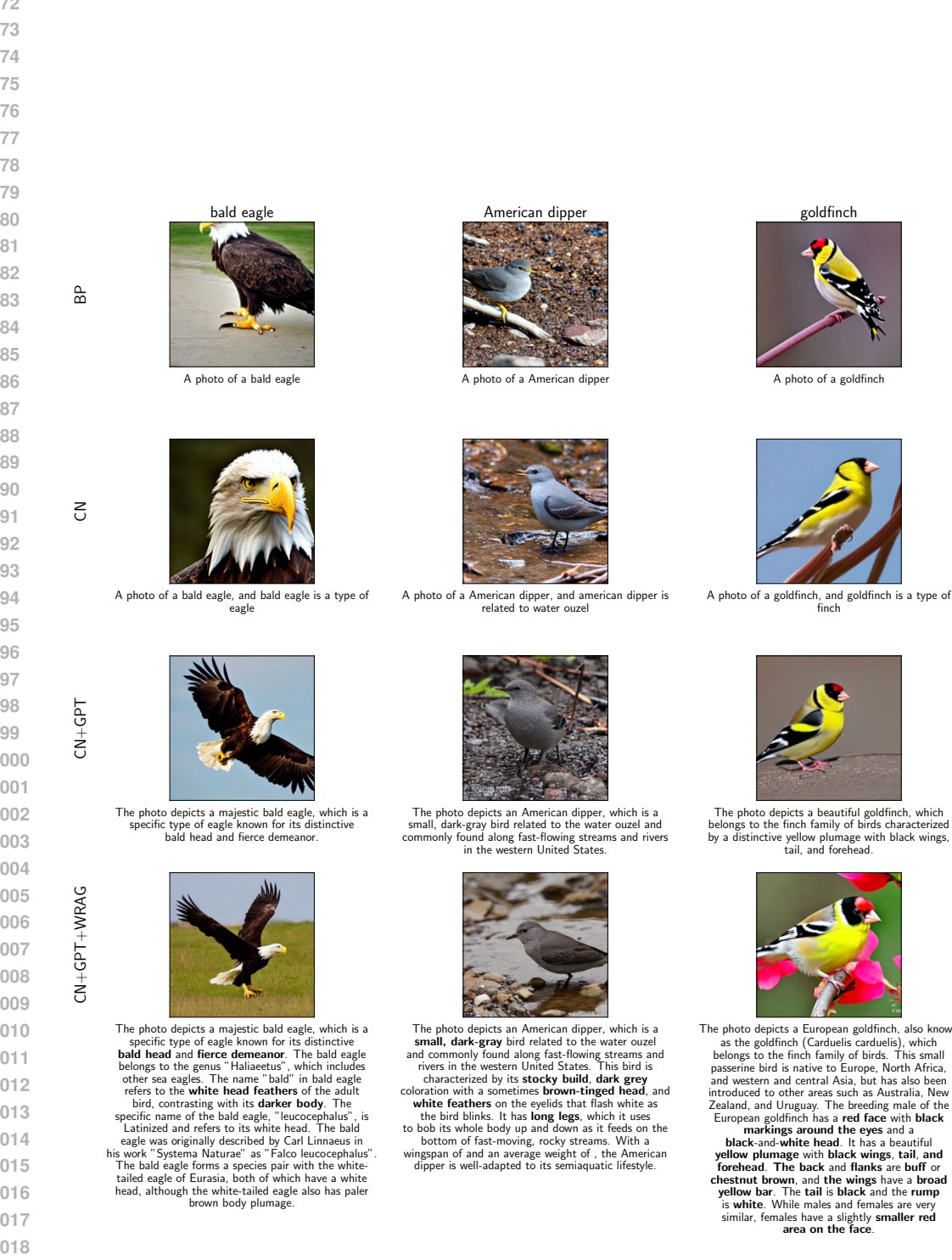

Figure 7: Examples of generated image-text pairs for birds.

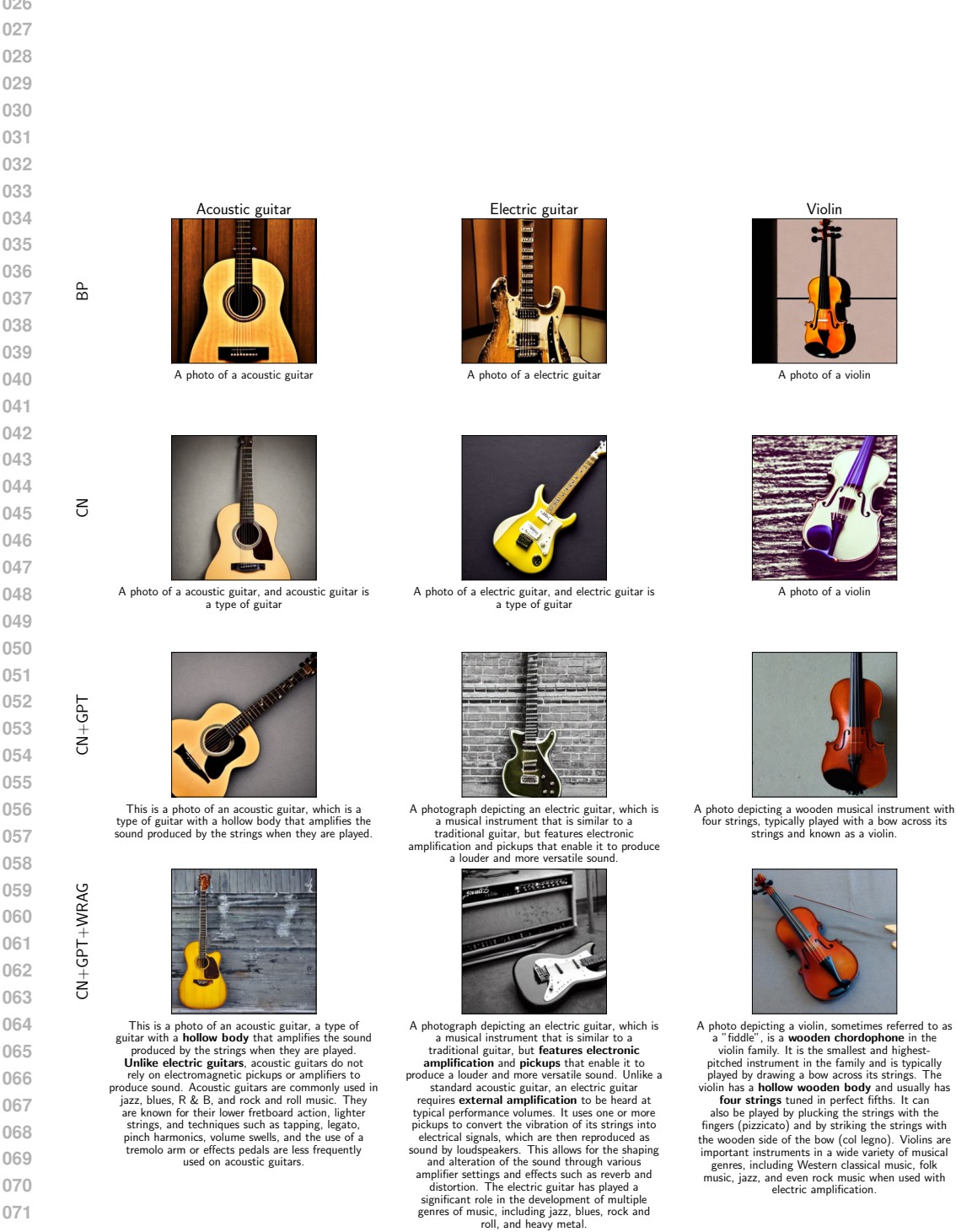

Figure 8: Examples of generated image-text pairs for non-animal objects.

