# OpenReview forum: "KnowData: Knowledge-Enabled Data Generation for Improving Multimodal Models"
_ICLR.cc/2025/Conference — Submitted to ICLR 2025_

### Official Review · Reviewer_qRxQ · 2024-10-27

**Soundness:** 2
**Presentation:** 2
**Contribution:** 3
**Rating:** 6
**Confidence:** 4

**Summary:**

The paper proposes a data augmentation method called KnowData, which leverages text-to-image generation models to synthesize text-image pairs. Compared to traditional data augmentation methods, this approach incorporates structured knowledge from ConceptNet, covering taxonomy, relationships, and attributes, as well as unstructured sources like Wikipedia and Retrieval Augmented Generation (RAG) to enhance contextual and background details. GPT-3.5 is used to generate coherent sentences, and a text-to-image generation model creates diverse images, from which high-quality images are selected. The resulting dataset is used to fine-tune models. Experimental results show that this data augmentation method significantly enhances CLIP’s performance on image classification tasks.

**Strengths:**

1. The proposed data augmentation method addresses the limitations of existing text-image pairs that contain only class names or superficial or short descriptions. By incorporating structure and unstructure knowledge from knowledge bases and knowledge graphs, the resulting text descriptions are rich and reliable, thereby enhancing CLIP’s perception and cognition. This finding will encourage further studies on synthetic data that integrates real-world knowledge from knowledge bases or knowledge graphs.
2. The paper conducts extensive experiments, demonstrating the method's advantages from various perspectives. This paper also provides valuable insights and experience that benefit the community’s exploration in this area.
3. The study validates an interesting insight: longer, knowledge-rich textual descriptions do not limit image diversity in image augmentation. Instead, they may increase it. This finding can offer confidence for similar applications in long-text-to-image data synthesis, particularly for entity-centric data.

**Weaknesses:**

1. The experimental setup is inaccurately categorized. Although the paper frequently mentions a focus on zero-shot image classification, the augmented textual descriptions begin with class names derived from the training set used in the evaluation tasks. As a result, the final augmented image-description pairs are essentially a fine-grained version of the training set. While the KnowData-enhanced CLIP model was not directly fine-tuned on the original training set, the augmented data is an extension of the training data, which does not meet the criteria for zero-shot learning.
2. The paper contains numerous writing issues, primarily with the overly broad title and unclear focus, as well as some inaccurate statements.  I hope you can revise it in the future. If the revision results in a version that is clearer and more accurate, it will deserve a score above the borderline.
3. Some citations are missing or inappropriate. While the authors have generally identified relevant papers, certain sources are incorrectly cited multiple times throughout the paper.

**Questions:**

I. Title and Focus are Inappropriate

-- 1. The paper’s focus is too broad, which leads to confusion about the specific model and task under study. For example, in the first paragraph, it may not be suitable to discuss both vision-language pre-trained models and diffusion models, like Flamingo and DALL-E 3, together. Flamingo is a vision-language pre-trained model, while DALL-E 3 is a diffusion model. The broad scope introduces unrelated topics, such as automatic content generation, auxiliary techniques, and image-text retrieval. It is suggested to narrow the focus to Contrastive Language-Image Pre-training (CLIP).

-- 2. The broad focus results in certain claims that cannot be sufficiently supported by experiments on CLIP alone, such as in lines 97-98, “enhancing the learning efficacy and application potential of multimodal models.” However, the study only tests CLIP without exploring other models.

-- 3. It is suggested that the title be narrowed down, for instance, to "Knowledge-Enabled Data Generation for Long-Tail Entity Image Classification," as focusing solely on class names, knowledge graphs, and Wikipedia descriptions with images is inadequate for constructing datasets for more complex tasks. Lines 96-97 mention “domain-specific insights into the text generation process,” yet “domain-specific” does not apply to data synthesis methods for tasks beyond classification.

-- 4. While Table 7 extends beyond image classification to VQA and image-text matching, it remains uncertain if such improvements are consistently present. KnowData’s generated text-image pairs are entity-centric and cater to less common entities that benefit from additional background knowledge. However, VQA and Winoground require inference and compositional generalization, which are not covered by entity image-description pairs. Perhaps you could further explain why domain-specific entity knowledge leads to improvements in VQA and compositional generalization tasks.

II. Citation Issues

-- 1. Due to the overly broad focus in the first two paragraphs of the introduction, related work citations are insufficient. Each category or application is cited with only one reference, which is not necessarily the latest or most representative.

-- 2. Line 44 cites Radford et al. (2021) as mentioning that CLIP uses noisy image-text pairs from the internet. However, this does not support the claim that traditional dataset collection methods compromise contextual richness and accuracy due to noisy internet data. Instead, Radford et al. (2021) demonstrates that CLIP’s effectiveness surpasses that of previous models using a large and noisy internet-based image-text dataset. It is suggested to add additional references on “contextual richness” and “accuracy” which are impacted by “noisy data”.

-- 3. The paper does not adequately explain the necessity and challenges of the proposed method. Lines 50–52 in the second paragraph should focus more on this, given the prevalence of synthetic data in image classification and other multimodal tasks. The paper only lists one related work, but several studies are relevant, such as Synthetic (He et al., 2023) and Diversity (Shipard et al., 2023). To strengthen credibility, more relevant works should be cited here. Additionally, while DALL-E 3’s caption construction method may have inspired this work, it does not effectively demonstrate the need for the proposed method.

-- 4. Line 189 references GPT-3.5 incorrectly with (Brown et al., 2020). (Brown et al., 2020) is related to GPT-3. The appropriate citation for GPT-3.5 is the paper of InstructGPT.

III. Writing Issues

-- 1. Line 110-111, “This study aims to set a new benchmark and ignite further research into knowledge-guided approaches for multimodal learning.” The paper presents a dataset construction process, which results in a training set, not a benchmark.

-- 2. Separate different conclusions. For instance, “Our extensive experiments demonstrate that CLIP models fine-tuned with our knowledge-guided, synthesized dataset outperform those trained with state-of-the-art (SOTA) data generation approaches or other zero-shot techniques,” represents two conclusions.

-- 3. Line 173-174, the mention of ATOMIC (Hwang et al., 2021) providing commonsense knowledge around human events seems unnecessary.

-- 4. Line 293 mentions “robustness and adaptability”; however, robustness was not evaluated, so “robustness” should be removed.

-- 5. Line 330-331, “P denotes the use of pre-trained models” should clarify which pre-trained models are meant. Is this referring to CLIP? If so, consider rephrasing to “P denotes the use of only CLIP.”

-- 6. Line 428, directly linking the appendix figures in the main text could be avoided; Figure 2 is sufficient, even though I reviewed figures 6 and 8 in the appendix.

-- 7. Line 469, the claim “KnowData enables better data scaling law” is typically about the trend in model performance with an increase in training data scale. However, for highlighting KnowData’s effectiveness over basic prompt usage, “KnowData utilizes data more efficiently” may be more appropriate.

IV. Inaccurate Statement of Experimental Setup

-- 1. The paper does not constitute a zero-shot setting. This paper says that “as no original training data is used in our framework, our evaluation belongs to the zero-shot setting.” However, in the experimental setup, “We generate 480k synthetic images based on ImageNet class names to fine-tune the downstream models and then evaluate the fine-tuned models on ImageNet test data and its out-of-distribution variants. We generate about 60k images for other datasets with fewer categories, including CIFAR100, DTD, and EuroSAT.” Although not the original training set, the similar training data and augmented descriptions (with labels) & images are used. Therefore, this setting does not meet the criteria for zero-shot. Similarly, the title “KnowData improves CLIP’s zero-shot performance” in line 368 is also inaccurate.

[Optional] V. Experimental Results Needing Further Explanation （Further details could enhance the story, but their absence is not critical）

-- 1. Why does the KnowData augmentation method underperform compared to the OpenAI-version CLIP on ImageNet-A? The paper does not explain this, so a definite or plausible explanation would be helpful.

-- 2. In text-to-image generation, DALL-E 3 often surpasses Stable Diffusion v1.5 in image quality. Yet, in Table 4, DALL-E 3 performs similarly to or worse than Stable Diffusion v1.5 on several datasets. This could result from the more realistic nature of Stable Diffusion v1.5 images or its closer alignment with test distribution on simple, single-object images. Additional explanations are expected.

-- 3. In Table 5, the CN+WRAG+GPT+Div setup greatly improves DTD and EuroSAT while offering a smaller boost for ImageNet. This may be domain-specific. Given the high training cost (140k image-text pairs), can the study identify which classes benefit more from knowledge and diversity enhanced prompts? Additionally, are there classes where basic prompts are sufficient?

-- 4. The effect of diversity trick (with different guidance scales) is unclear. “Diversity in knowledge and diversity in images both matter” in Line 482 lacks direct experimental evidence. Table 5 does not include a diversity score. Is it possible that CN+WRAG+GPT alone (without the diversity trick) could improve diversity scores?

---

> ### Author Response · Authors · 2024-11-24
>
> > I. Title and Focus are Inappropriate (1-3)
>
> Answer for 1-3: Thanks for the insight comment. Following your suggestion, we revised the statement to focus on CLIP.  Our main contribution is to provide a general multi-modal data generation framework. To evaluate such data, we focused on improving CLIP models as one application.
>
> We will revise the title to “KnowData: Knowledge-Enabled Multimodal Data Generation for Long-Tail Entity Image Recognition,” as suggested in a future version.
>
>
> >4. While Table 7 extends beyond image classification to VQA and image-text matching, it remains uncertain if such improvements are consistently present. KnowData’s generated text-image pairs are entity-centric and cater to less common entities that benefit from additional background knowledge. However, VQA and Winoground require inference and compositional generalization, which are not covered by entity image-description pairs. Perhaps you could further explain why domain-specific entity knowledge leads to improvements in VQA and compositional generalization tasks.
>
> Answer for 4:
> In the experiments presented in Table 7, KnowData generates synthetic multimodal data for 1,000 ImageNet classes, encompassing general knowledge about natural objects. This general knowledge may overlap with some of the information required to address the VQA and Winoground tasks. Another potential explanation is that the fine-tuned encoders trained on KnowData's synthetic data are better equipped to distinguish similar image-text pairs. This capability arises from the detailed and nuanced knowledge descriptions provided by the synthetic data, which help the model capture subtle differences. These enhanced representations may contribute to improved performance in tasks requiring compositional generalization and inference.
>
> > II. Citation Issues
>
> Thanks so much for the detailed feedback. We have revised the related work and citations accordingly.
>
> > III. Writing Issues
> >1. Line 110-111, “This study aims to set a new benchmark and ignite further research into knowledge-guided approaches for multimodal learning.” The paper presents a dataset construction process, which results in a training set, not a benchmark.
>
> A: Thanks for the suggestion. We have revised the statement to focus on the data generation process.
>
> >2. Separate different conclusions. For instance, “Our extensive experiments demonstrate that CLIP models fine-tuned with our knowledge-guided, synthesized dataset outperform those trained with state-of-the-art (SOTA) data generation approaches or other zero-shot techniques,” represents two conclusions.
>
> A: Thanks for the suggestion. We have revised the statement to separate these two parts.
>
>
> >3. Line 173-174, the mention of ATOMIC (Hwang et al., 2021) providing commonsense knowledge around human events seems unnecessary.
>
> A: Thanks for the suggestion. This demonstrates the reason why we chose ConceptNet, as it better aligns with the knowledge domain of the data that we aim to generate in the subsequent sections.
>
> >4. Line 293 mentions “robustness and adaptability”; however, robustness was not evaluated, so “robustness” should be removed.
>
> A: We tested the accuracy on ImageNet-related datasets, and the overall improvement (IN-avg) demonstrates the out-of-distribution robustness.
>
>
> >5. Line 330-331, “P denotes the use of pre-trained models” should clarify which pre-trained models are meant. Is this referring to CLIP? If so, consider rephrasing to “P denotes the use of only CLIP.”
>
> A: Yes, "P" denotes the use of pre-trained CLIP in those cases, others may train CLIP from scratch for example.
>
>
> >6. Line 428, directly linking the appendix figures in the main text could be avoided; Figure 2 is sufficient, even though I reviewed figures 6 and 8 in the appendix.
>
> A: Thanks for the suggestion. We have fixed them in the revision.
>
> >7. Line 469, the claim “KnowData enables better data scaling law” is typically about the trend in model performance with an increase in training data scale. However, for highlighting KnowData’s effectiveness over basic prompt usage, “KnowData utilizes data more efficiently” may be more appropriate.
>
> A: Thanks for the suggestion. In Figure 3, we illustrate the trend of model performance as the scale of synthetic data increases, aligning with the reviewer’s observation regarding scaling laws. We have revised the conclusion to reflect that “KnowData utilizes data more efficiently”.

---

> ### Author Response · Authors · 2024-11-24
>
> > IV. Inaccurate Statement of Experimental Setup
>
> >1. The paper does not constitute a zero-shot setting. This paper says that “as no original training data is used in our framework, our evaluation belongs to the zero-shot setting.” However, in the experimental setup, “We generate 480k synthetic images based on ImageNet class names to fine-tune the downstream models and then evaluate the fine-tuned models on ImageNet test data and its out-of-distribution variants. We generate about 60k images for other datasets with fewer categories, including CIFAR100, DTD, and EuroSAT.” Although not the original training set, the similar training data and augmented descriptions (with labels) & images are used. Therefore, this setting does not meet the criteria for zero-shot. Similarly, the title “KnowData improves CLIP’s zero-shot performance” in line 368 is also inaccurate.
>
> A: Thank you for your valuable feedback. We would like to clarify that in our study, we only use the class label names to generate the data. Therefore, no similar training data, augmented descriptions, or images are used in our pipeline. Moreover, we follow prior established work, such as He et al.[1] , which also considers this approach as a zero-shot setting using only the class names. We believe this methodology aligns with current standards in the field.
>
> - [1] Is synthetic data from generative models ready for image recognition? ICLR 2022
>
> > [Optional] V. Experimental Results Needing Further Explanation （Further details could enhance the story, but their absence is not critical）
>
> >1. Why does the KnowData augmentation method underperform compared to the OpenAI-version CLIP on ImageNet-A? The paper does not explain this, so a definite or plausible explanation would be helpful.
>
> A: Thank you for your insightful question. The underperformance of the KnowData augmentation method compared to the OpenAI-version of CLIP on ImageNet-A could be attributed to several factors.
>
> Firstly, ImageNet-A is a challenging dataset composed of adversarial and hard-to-classify images that are specifically designed to mislead models, and may have a distribution shift. The OpenAI-version of CLIP has been extensively trained on a vast and diverse dataset, which may include images similar to those in ImageNet-A, giving it an inherent advantage.
>
> Secondly, our KnowData method relies solely on class label names to generate data, without utilizing similar training data, augmented descriptions, or additional images in the pipeline. This minimalist approach, may not provide enough contextual information for the model to accurately classify the complex and unusual images found in ImageNet-A.
>
> We acknowledge that the paper does not elaborate on this aspect, and we appreciate your feedback. In future work, we plan to investigate this performance gap further and explore enhancements to our augmentation method, potentially by incorporating additional contextual information or by testing on more advanced multimodal models to improve results on challenging datasets like ImageNet-A.
>
> > 2. In text-to-image generation, DALL-E 3 often surpasses Stable Diffusion v1.5 in image quality. Yet, in Table 4, DALL-E 3 performs similarly to or worse than Stable Diffusion v1.5 on several datasets. This could result from the more realistic nature of Stable Diffusion v1.5 images or its closer alignment with test distribution on simple, single-object images. Additional explanations are expected.
>
> A: Thank you for your insightful question. Apart from the reason you state, one possible explanation for this observation is that, despite the superior image quality of DALL-E 3, the level of knowledge embedded within both models may be similar. In zero-shot classification tasks, the amount of domain-specific knowledge that a model can convey through generated images plays a crucial role. If both models encode and express similar levels of knowledge about the classes, they might achieve comparable performance in classification, regardless of differences in image quality.

---

> ### Author Response · Authors · 2024-11-24
>
> >3. In Table 5, the CN+WRAG+GPT+Div setup greatly improves DTD and EuroSAT while offering a smaller boost for ImageNet. This may be domain-specific. Given the high training cost (140k image-text pairs), can the study identify which classes benefit more from knowledge and diversity enhanced prompts? Additionally, are there classes where basic prompts are sufficient?
>
> A: Thanks for your valuable advice! Based on your suggestions, we created two figures to describe the degree to which each class benefits from knowledge and diversity (i.e., the accuracy improvement before and after fine-tuning). To determine whether the base prompts are sufficient, we also plotted the initial pretrained accuracy of these classes together. From [the top 20 labels with the highest accuracy improvements]( https://i.ibb.co/hdWLZX8/ACC-top20-improve-imagenet-vitb16.png ) and [the overall LOWESS trends]( https://i.ibb.co/fS9KRvr/ACC-improve-imagenet-vitb16.png ), it is evident that classes with initially low accuracy, where base prompts are less sufficient, benefit more from knowdata. These classes may not have enough knowledge to distinguish similar classes, such as 'hare' and 'sailboat' in the top 20, and the knowledge from Knowdata can supplement the base prompts.
>
>
> >4. The effect of diversity trick (with different guidance scales) is unclear. “Diversity in knowledge and diversity in images both matter” in Line 482 lacks direct experimental evidence. Table 5 does not include a diversity score. Is it possible that CN+WRAG+GPT alone (without the diversity trick) could improve diversity scores?
>
> A: Thanks for your valuable advice! Firstly, the improvement of 'BP+Div' over 'BP' in Table 5 has already demonstrated the importance of the diversity trick. Secondly, we have added experiments here on controlling the diversity/guidance scale with CN+WRAG+GPT as follows:
>
> |Method|IN-Val|IN-V2|IN-R|IN-A|IN-Sketch|ObjectNet|Average|
> | :--: |:--: |:--: |:--: |:--: |:--: |:--: |:--: |
> |Knowdata|69.95|63.61|78.18|48.81|49.93|55.36|60.97|
> |Control diversity|69.73|62.90|77.44|50.07|49.02|54.68|60.64|
>
> As can be seen, even in CN+WRAG+GPT, where the prompts are already quite rich and possess a considerable degree of diversity, our diversity trick remains effective.

---

> ### Author Response · Authors · 2024-12-02
>
> Dear Reviewer,
>
> Thank you once again for your detailed comments and suggestions. As the rebuttal period is approaching its end, we would greatly appreciate your feedback on whether our responses and revision have addressed your concerns. We are also happy to engage in further discussions if needed.

---

> ### Comment · Reviewer_qRxQ · 2024-12-02
>
> I appreciate the response and thorough clarifications provided. The response to question I-4 was particularly illuminating, especially regarding how "This capability arises from the detailed and nuanced knowledge descriptions provided by the synthetic data, which help the model capture subtle differences."
>
> The two additional figures (top 20 labels with highest accuracy improvements and LOWESS trends) are valuable additions that strengthen the paper's conclusions. I recommend including them in the appendix as further discussion.
>
> Based on these revisions, I will raise the score to 6.
>
> One technical question remains: Could you clarify whether the x-axis represents steps, epochs, or data size? Also, regarding the LOWESS trends, does the figure indicate consistent model improvement after the initial 100 units, as suggested by the positive improvement values?

---

> > ### Author Response · Authors · 2024-12-03
> >
> > A: Thank you again for your valuable feedback and positive rating. Your support is greatly appreciated.
> >
> > To address your point, the x-axis in the LOWESS trends represents the 1,000 classes in ImageNet. The reviewer is correct that aside from the initial 100 classes, the remaining classes generally show significant model improvement. We believe this is because classes with lower pre-trained accuracy benefit more from the knowledge in our synthetic data. On the other hand, for classes that already have high pre-trained accuracy, fine-tuning on synthetic data may suffer from the distribution shift between synthetic and real data, which can limit the effectiveness of the knowledge-enabled synthetic data.
> >
> > We will include two additional figures, Top 20 Labels with Highest Accuracy Improvements and LOWESS Trends, in our revised manuscript to provide further discussion and support for the paper's conclusions.
> > Thank you once again for your constructive comments and continued support.

---

> > > ### Comment · Reviewer_qRxQ · 2024-12-03
> > >
> > > Thank you for your thorough response. I truly appreciate how this work demonstrates the effectiveness of synthetic data generation based on multiple knowledge graphs, showing clear benefits through detailed knowledge descriptions.
> > >
> > > Looking forward to seeing this line of research develop further!

---

> > > > ### Author Response · Authors · 2024-12-03
> > > >
> > > > Thank you again for the valuable feedback!

---

### Official Review · Reviewer_oNom · 2024-11-03

**Soundness:** 2
**Presentation:** 3
**Contribution:** 2
**Rating:** 5
**Confidence:** 4

**Summary:**

This work proposes a data augmentation methodology called KnowData. It basically consists of four main components/procedures: (1) structured knowledge generation to obtain schema-based knowledge, (2) LLM-aided summarization and refinement for the previously obtained structured knowledge, (3) generating images from the crafted/refined prompts from step (2), and finally (4) finetuning multimodal models from the augmented data obtained from step 1 to 3.
The authors then demonstrate the effectiveness of the proposed method on numerous datasets compared to baselines such as the vanilla CLIP and some of its zero-shot enhanced variants.

**Strengths:**

- The paper is quite clearly written, easy to follow, and well-illustrated.
- The proposed data augmentation method improves the zero-shot performance quite significantly, where comprehensive experimental results are shown.
- The retrieve and summarize idea is neat.

**Weaknesses:**

- The CLIP score is a holistic score, what about finer-granularity of details that are suboptimal but decisive for recognition?
- The proposed data-augmentation method is supposedly generalizable, yet the paper only tests its effectiveness on one multimodal model, that is CLIP. The method should be further justified by more advanced multimodal LLM-based models in a controlled (few-shot) setting.
- The proposed method is rather incremental, and there is no direct evaluation applicable (in the paper) to quantify how and where the generated prompts are better than normally collected ones. Ablations needed compared to web-crawled image captions in addition to self-ablations in Table 5.
- The proposed method needs to be tested on datasets that are much more challenging, especially ones that contain sufficiently confusing class-pairs, such as iNaturalist (Horn et al.).

**Questions:**

- What is the performance between confusing pairs? I.e., when we plot the confusion matrix, and select the top-2 (or K) confusing class-pairs, how much does the proposed model improve?

---

> ### Author Response · Authors · 2024-11-24
>
> Thank you for your valuable suggestions!
>
> >Q1: The CLIP score is a holistic score, what about finer-granularity of details that are suboptimal but decisive for recognition?
>
> A1: Thank you for the valuable comment.  We would like to clarify that our approach is justified for several reasons:
> 1. **Purpose of filtering**: Our primary goal in using CLIP scores is not to perform precise quality ranking, but rather to eliminate obviously mismatched samples or failed generations for the targeted class. As noted in lines 259-267 of our paper, we are targeting clear failure cases rather than making fine-grained quality distinctions.
> 2. **knowledge enables finer-granulartiy of details**: In Figures 2 and 6 to 8, we display pairs of image-text from different numbers of knowledge sources and annotate helpful information in the prompt from each knowledge source. It can be find that knowledge-enabled image generation can present more accurate details to help differentiate similar classes. Take Figure 2 with three similar dog species (Weimaraner, Rhodesian Ridgeback, and Vizsla) as an example. We see that with the base prompt, the generated images can distinguish Weimaraner but cannot differentiate between Rhodesian Ridgeback and Vizsla. However, with the addition of knowledge, the generated images can differentiate them through the distinct coat colors (Rhodesian Ridgeback with light wheaten or red wheaten coat, and Vizsla with rust-colored coat) and the unique nose color of Vizsla (reddish-colored nose, which blends with their coat color).
>
> >Q2: The proposed data-augmentation method is supposedly generalizable, yet the paper only tests its effectiveness on one multimodal model, that is CLIP. The method should be further justified by more advanced multimodal LLM-based models in a controlled (few-shot) setting.
>
> Thank you for your valuable feedback. We acknowledge that although our proposed data augmentation method is intended to be generalizable, we have only tested its effectiveness on one multimodal model. While our current work does not focus on this aspect, we included a discussion of its potential as a future research direction in Appendix B of our revision.
> We believe this will help to further strengthen our research findings.
>
> >Q3: The proposed method is rather incremental, and there is no direct evaluation applicable (in the paper) to quantify how and where the generated prompts are better than normally collected ones. Ablations needed compared to web-crawled image captions in addition to self-ablations in Table 5.
>
> Thank you for the suggestion. We acknowledge that crawling high-quality web image captions and selecting the most relevant ones for each class label is a challenging and non-trivial task, which could itself constitute a significant contribution. While our current work does not focus on this aspect, we included a discussion of its potential as a future research direction in Appendix B of our revision.
>
> >Q4: The proposed method needs to be tested on datasets that are much more challenging, especially ones that contain sufficiently confusing class-pairs, such as iNaturalist (Horn et al.).
>
> Thank you for the suggestion. In Table 7, we evaluate our method on WinoGround, which is a similarly confusing and challenging dataset, and observe improved performance from KnowData. This demonstrates the robustness of our approach to handling complex and ambiguous cases. We appreciate the recommendation of iNaturalist and will include it in our future work.
>
> >Q5: What is the performance between confusing pairs? I.e., when we plot the confusion matrix, and select the top-2 (or K) confusing class-pairs, how much does the proposed model improve?
>
> A5: Thanks for your valuable advice! According to your suggestions, we draw confusion matrix on CLIP RN50 with/without finetuned on Knowdata. Considering for the clear visualization, we choose Cifar100 which has suitble label numbers. You can see the comparison of two matrix in [this anoymous link]( https://i.ibb.co/7YCGBHX/cifar100-rn50-confmatrix.png ). It can be seen that after finetuning, there are fewer misclassified labels.
>
> In addition, we selected the top two most confusing class pairs and calculated the improvements for these pairs before and after fine-tuning. Specifically:
>
> - **The most confused class pair is**: the true class **"orchid"** being incorrectly predicted as **"tulip"**.
>  - The number of confusions decreased from **52.0** to **16.0**, an improvement of **36.0**, which is a **69.23%** reduction.
>  - **The second most confused class pair is**: the true class **"ray"** being incorrectly predicted as **"flatfish"**.
>  - The number of confusions decreased from **45.0** to **16.0**, an improvement of **29.0**, which is a **64.44%** reduction.

---

> > ### Comment · Reviewer_oNom · 2024-11-29
> >
> > Thanks the authors for the responses.
> > Some did acknowledged my questions/concerns, however, overall I suggest the paper at its current version has quite obvious rooms for improvements.
> > For those points mentioned left as future work, I actually do think they're required to make this work sound and impactful.
> > The work here aimed at proposing a generic framework but the most straightforward way to verify that confidently is through more empirical evidences.
> > I do hope to see a much more improved version of the work in the future venue submission.

---

> > > ### Author Response · Authors · 2024-12-02
> > > **Response to Reviewer oNom**
> > >
> > > We greatly appreciate your acknowledgment of the aspects we addressed in response to your questions and concerns. We are committed to incorporating your suggestions and will focus on expanding experimental validation in revision. Thank you again for your comments.

---

### Official Review · Reviewer_6gNC · 2024-11-03

**Soundness:** 3
**Presentation:** 3
**Contribution:** 2
**Rating:** 5
**Confidence:** 4

**Summary:**

The paper proposes a new technique to improve the zero-shot classification performance of vision-language models by incorporating factual knowledge data for the class descriptions. Specifically, KnowData starts from the class names of an image classification dataset, enriches them with ConceptNet relations, GPT-3.5, and RAG-retrieved Wikipedia knowledge, to create several detailed descriptions. These descriptions are then used to generate synthetic images with a text-to-image diffusion model. Finally, the synthetic dataset is used to fine-tune a vision-language model on the dataset-specific synthetic data.

**Strengths:**

- Incorporating facts from knowledge bases such as ConceptNet and Wikipedia is a reasonable approach to ensure correctness while obtaining a diverse set of novel class descriptions.
- The zero-shot classification results improve over previous works, especially for fine-grained classification datasets, such as DTD and EuroSAT.
- A comprehensive set of ablations studies evaluate the individual parts of KnowData, and further analyze the effect of the diffusion model, the scaling ability, and the VLM size.
- Zero-shot performance of the fine-tuned model on other downstream tasks (VQAv2, WinoGround) suggest that KnowData generalizes beyond image classification.

**Weaknesses:**

- Improvements on ImageNet, although better than previous work, are rather marginal.
- The selection of fine-grained datasets is rather slim and could be extended. Cited works such as [He et al., 2023] evaluated on many more datasets.
- The CLIP fine-tuning is not entirely clear. Is a randomly initialized classification head added on top of the CLIP vision features, i.e. linear probing, or are the text embeddings of the base prompt used for classification during fine-tuning? If it is the former, how does the generalization in Tab. 7 work, i.e., how does KnowData ensure that the fine-tuned model is still compatible with the text encoder embeddings?
- Some additional ablations could help better understand all components of the method:
  * With vs. without few-shot demonstrations for Sec. 3.2 description summarization and refinement.
  * With vs. without CLIP filtering in Sec. 3.3 image generation.
  * Training only a classification head vs. multiple layers in addition. This would create a fair comparison to "Synthetic".
- Tab. 7 already suggests this generalizes beyond the specific use case of image classification. This paper would have been more impactful, if it fully embraced this direction to improve all vision-language capabilities of CLIP, e.g. text/image retrieval, instead of focusing mainly on image classification.
- While the idea to incorporate factual knowledge is novel, it is a rather incremental improvements over previous approaches that also used external knowledge. I see this as a minor weakness.

**Questions:**

Please address the points raised in the Weaknesses section. Here are additional relevant questions/comments:
- L. 185: How do you choose the N structured knowledge descriptions when there are more than N? What do you do when there are less than N?
- Tab. 3 shows CLIP scores, but it is not clear which text embedding is used for each row. A fair comparison would use the base prompt for each row. Please clarify.

---

> ### Author Response · Authors · 2024-11-24
>
> Thank you for your valuable suggestions!
> >Q1: Improvements on ImageNet, although better than previous work, are rather marginal.
>
> A1: Thanks for the great question!
>
> Firstly, the reason for minor improvements on some datasets, such as ImageNet, is due to their challenging nature. **The SOTA baseline, Synthetic [4], under the CLIP ViT-B/16 architecture, only achieved a +0.56 improvement in accuracy relative to the OpenAI CLIP on ImageNet, whereas our improvement is +1.84**.
>
> Moreover, it is worth noting that our baselines are quite strong. In Table 1, our competitors ZPE [1], Description [2], and Hierarchy [3] compared with baselines that use the class name as the initial text embedding for a zero-shot CLIP. The Synthetic [4] compared with baselines that use an OpenAI template as the initial text embedding for a zero-shot CLIP, and while Diversity [5] indeed uses Synthetic [4] as its baseline, it only compares small datasets such as CIFAR-10, CIFAR-100, and EuroSAT. In contrast, we compare against the best among them for every dataset and have conducted comparisons across a large-scale dataset (ImageNet), fine-grained datasets (DTD, EuroSAT), and robustness (4 ImageNet variant test datasets). Therefore, our ability to surpass the SOTA with our baseline indicates that we have made significant improvements over simpler baselines.
>
> - [1] A simple zero-shot prompt weighting technique to improve prompt ensembling in text-image models. ICML 2023
> - [2] Visual classification via description from large language models. ICLR 2023
> - [3] Improving zero-shot generalization and robustness of multi-modal models. CVPR 2023
> - [4] Is synthetic data from generative models ready for image recognition? ICLR 2022
> - [5]  Diversity is definitely needed: Improving model-agnostic zero-shot classification via stable diffusion. CVPR 2023
>
> >Q2: The selection of fine-grained datasets is rather slim and could be extended. Cited works such as [He et al., 2023] evaluated on many more datasets.
>
> A2: Thanks for your valuable suggestions. We clarify that in our experiments, we use representative datasets across different domains:
> - (1) object-level datasets: ImageNet, CIFAR 100 for natural images,
> - (2) fine-grained dataset: EuroSAT for satellite and land images, and DCD for textural images in the wild
> - (3) robustness datasets: 5 ImageNet out-of-distribution robustness datasets.
>
> We will include more fine-grained datasets in the future to further demonstrate the general applicability of KnowData.
>
> >Q3: The CLIP fine-tuning is not entirely clear. Is a randomly initialized classification head added on top of the CLIP vision features, i.e. linear probing, or are the text embeddings of the base prompt used for classification during fine-tuning? If it is the former, how does the generalization in Tab. 7 work, i.e., how does KnowData ensure that the fine-tuned model is still compatible with the text encoder embeddings?
>
> A3: Thanks for your valuable suggestions. The classification head is initialized by the standard [OpenAI prompt template](https://github.com/openai/CLIP/blob/main/data/prompts.md) used for classification.  And the details of CLIP finetuning are discussed in 279-288 lines; we achieve better results by fine-tuning part of the pre-trained image encoder parameters in addition to the classification head.

---

> > ### Author Response · Authors · 2024-11-24
> >
> > >Q4: Some additional ablations could help better understand all components of the method: 1) With vs. without few-shot demonstrations for Sec. 3.2 description summarization and refinement. 2) With vs. without CLIP filtering in Sec. 3.3 image generation. 3) Training only a classification head vs. multiple layers in addition. This would create a fair comparison to "Synthetic".
> >
> > A4:
> > 1) The comparison of With vs. without few-shot demonstrations for Sec. 3.2 description summarization and refinement can be seen in Appendix D.
> > 2) The quantity comparison of With vs. without few-shot demonstrations for Sec. 3.2 description summarization and refinement can be seen in Appendix
> > F (Figure 4& Figure5), here we add quality comparison between different CLIP filtering rates `\theta=  0%, 20%, 40%, 60%`:
> >
> >   | theta (CLIP score filter out rate) | DTD results on ViT-B/16 |
> >   | :--: | :--: |
> >   | 0% | 56.38 |
> >   | 20% (KnowData) | **57.51** |
> >   | 40% | 55.31 |
> >   | 60% | 53.07 |
> >
> >    These results demonstrate that: a) Using no filtering ($\theta=0$) incorporates low-quality samples, slightly hurting performance. b) Aggressive filtering ($\theta=40%, 60%$) reduces training data too much, leading to decreased performance. c) Our chosen rate  ($\theta=20%$) provides the best performance, suggesting it effectively removes obviously problematic samples while retaining sufficient training data.  We intentionally chose a moderate filtering ratio (20%) to balance removing obvious errors with retaining a diverse dataset.
> >
> > 3) Here we add comparison on OpenAI CLIP, Synthetic, Training only a classifcation head with Knowdata and Training multiple layers with Knowdata:
> >
> > | Model | Method | CIFAR100 | DTD | EuroSAT |IN-Avg|
> > | :--: |:--: |:--: |:--: |:--: |:--: |
> > |CLIP RN50 | OpenAI | 41.60 | 41.70 | 41.10 | 46.25|
> > || Synthetic| 45.69 | 43.19 | 55.37 | 45.95 |
> > | | Training only a classifcation head with Knowdata| 49.60 | 52.13 | 57.20 | 47.01|
> > || Training multiple layers with Knowdata | 57.16 | 51.18 | 57.19 | 46.91 |
> > |CLIP ViT-B/16 | OpenAI | 68.70 | 46.00 | 54.10 | 61.25|
> > || Synthetic| 70.71 | 44.92 | 59.86 | 60.71 |
> > | | Training only a classifcation head with Knowdata| 70.86 | 55.91 | 59.84 | 61.78|
> > || Training multiple layers with Knowdata | 73.88 | 57.51 | 63.86 | 62.41|
> >
> > >Q5: Tab. 7 already suggests this generalizes beyond the specific use case of image classification. This paper would have been more impactful, if it fully embraced this direction to improve all vision-language capabilities of CLIP, e.g. text/image retrieval, instead of focusing mainly on image classification.
> >
> > Thank you for the valuable suggestion. We agree that extending our approach to improve all vision-language capabilities of CLIP, including text/image retrieval, is an important and exciting direction. While our current work primarily focuses on image classification, we have included a discussion of extending our methods to tasks like text/image retrieval in our future work section Appendix B."
> >
> > >Q6: While the idea to incorporate factual knowledge is novel, it is a rather incremental improvement over previous approaches that also used external knowledge. I see this as a minor weakness.
> >
> > Thank you for the valuable suggestion. Firstly, the idea of introducing knowledge into data generation and model training is novel. While leveraging synthetic data for training large-scale models is not uncommon, our paper presents a novel method to incorporate world knowledge from various knowledge sources including structured and unstructured knowledge bases.
> >
> > Secondly, the pipeline and the synthetic data are also our contributions. We have carefully designed the pipeline, and the generated text and images are of high quality, as evidenced by both the qualitative visualizations and the performance gains observed. The pipeline and the synthetic data could be useful for other applications such as improving downstream multimodal models
> >
> > Lastly, integrating diverse real-world knowledge into image-text pairs is an important problem. The way we leverage generative models may inspire other resource-constrained domains or revolutionize data annotation (moving from just annotating to generating).
> >
> > Based on reasons above, we think it is not a rather incremental improvement over previous approaches.

---

> > > ### Author Response · Authors · 2024-11-24
> > >
> > > >Q7: L. 185: How do you choose the N structured knowledge descriptions when there are more than N? What do you do when there are less than N?
> > >
> > > A7: When the knowledge descriptions is more than N, we will select the most relevant N descriptions, where in ConceptNet we use their inner weight parameter to judge similarity and in RAG we use KNN to compute embedding similarity. When the knowledge is less than N, we will randomly select one descriptions and repeat.
> > >
> > > To illustrate how is the N, including `N_conceptnet` and `N_rag` determined, We conducted additional experiments on the EuroSAT dataset using the ViT-B/16 CLIP model with `N_rag` values of 1, 2, and 3. In KnowData, by default, EuroSAT retrieves the top three relevant passages with $N_{rag}=3$ (as indicated in Table 1).
> > >
> > >   | N_rag | EuroSAT results on ViT-B/16 |
> > >   | :--: | :--: |
> > >   | 3 (KnowData) | **63.86** |
> > >   | 2 | 63.37 |
> > >   | 1 | 61.48 |
> > >
> > >   As we selected the top relevant passages, introducing useful information with increasing `N_rag` values leads to rapid performance improvements from $N_{rag}=1$ to $N_{rag}=2$.  From  $N_{rag}=2$ to $N_{rag}=3$, we still can observe improved accuracy but it is less apparent, which could be because the relevance of the added passages diminishes as we use larger $N_{rag}$. Here we select $N_{rag}=3$  as the default to balance effectiveness and computational efficiency. The progress is similar in `N_conceptnet`.
> > >
> > > >Q8: Tab. 3 shows CLIP scores, but it is not clear which text embedding is used for each row. A fair comparison would use the base prompt for each row. Please clarify.
> > >
> > >
> > > A8: Thank you for pointing this out. We clarify that for each method in Table 3, the CLIP scores are computed using the corresponding prompts and the generated images specific to that method. This approach ensures that the text-image pairs are appropriately aligned for a fair comparison. Using the base prompt for each row may not be entirely fair because different methods incorporate additional knowledge into their prompts, which influences the generated images and would not be accurately captured by the base prompt alone.

---

> > > > ### Comment · Reviewer_6gNC · 2024-11-29
> > > >
> > > > I would like to thank the authors for their response to my review and providing the additional experiments.
> > > >
> > > > I have some clarifying questions:
> > > > - Q3: Since it is now clear that you fine-tune a classifier head, I would like to know the answer to the second part of my question: How does the generalization in Tab. 7 work, i.e., how does KnowData ensure that the fine-tuned model is still compatible with the text encoder embeddings?
> > > > - Q4: Did you provide qualitative results with vs. without few-shot demonstrations (Sec. 3.2)? The answer sounds like you did, but I cannot find them.
> > > > - Q8: I disagree. There might be a bias of more comprehensive prompts leading to higher CLIP scores compared to simple prompts simply because they are less ambiguous about the image content (up to the point at which CLIP and the SD model can represent the text information accurately). Since the goal is to generate relevant data for a specific class, I believe the base prompt is the best proxy for this evaluation. If the authors disagree, then CLIP score might not be a good metric for what this evaluation is trying to show. I would argue it is not clear/proven that these values are comparable.

---

> ### Author Response · Authors · 2024-12-02
>
> >**Q3:** Since it is now clear that you fine-tune a classifier head, I would like to know the answer to the second part of my question: How does the generalization in Tab. 7 work, i.e., how does KnowData ensure that the fine-tuned model is still compatible with the text encoder embeddings?
>
> **A3:** Thank you for your question. For the Winoground test, we only use the image encoder fine tuned by synthetic ImageNet data generated by KnowData, as shown in Table 1. And we do not use the classification head of the ImageNet model. Instead, we leverage the off-the-shelf OpenAI CLIP text encoder to obtain text embeddings from the Winoground dataset captions. These text embeddings are then used to compute classification accuracy. The key idea is that the fine-tuned image encoder is capable of transferring its learned knowledge to other tasks, ensuring that the model remains generalizable across different datasets. This knowledge-enabled fine-tuning mechanism thus facilitates the generalization of the model, allowing it to effectively handle a variety of tasks.
>
> ---
>
> >**Q4:** Did you provide qualitative results with vs. without few-shot demonstrations (Sec. 3.2)? The answer sounds like you did, but I cannot find them.
>
> **A4:** Thank you for your question. In Appendix D, we provide a qualitative analysis highlighting the benefits of using few-shot demonstrations. Specifically, we show that without few-shot demonstrations, the language model may produce irrelevant outputs, such as "This sentence accurately describes a goldfish" (lines 800-802), rather than directly describing the image as shown in lines 786-788. By leveraging the in-context learning capability of the language model, we can design prompts that guide the model to provide more accurate and direct descriptions of the images. Additionally, within the few-shot demonstrations, we include some failure cases. These failures help the model respond as concisely as possible to irrelevant context, using keywords like "irrelevant" or "no information." This helps us quickly identify and exclude these failed cases in future iterations. We will add further clarification and provide more detailed results in the revised version of the paper.
>
> ----
>
>
> >**Q8:** I disagree. There might be a bias of more comprehensive prompts leading to higher CLIP scores compared to simpler prompts, simply because they are less ambiguous about the image content (up to the point at which CLIP and the SD model can represent the text information accurately). Since the goal is to generate relevant data for a specific class, I believe the base prompt is the best proxy for this evaluation. If you disagree, I think CLIP scores may not be the best metric for evaluating this task. It is still unclear or unproven that these values are directly comparable.
>
> **A8:** Thank you for your feedback. We would like to note that using the corresponding text and image pairs can reflect that the generated image faithfully present the knowledge in the prompts, which can not be represented by the simple class name (which can be ambiguous)
> Moreover, we demonstrate the superiority of our results primarily through the quantitative explanation in Table 1 (zero-shot performance) and the qualitative explanation in Figure 2. We acknowledge that CLIP scores may have some limitations, and we will make it clear in the updated version of the paper.

---

> > ### Comment · Reviewer_6gNC · 2024-12-02
> >
> > I would like to thank the authors for the additional clarifications.
> >
> > It is interesting and a bit surprising that the fine-tuned image encoder of of KnowData is compatible with the original CLIP text encoder.
> > I meant to ask whether there are quantitative evaluations with vs. without few-shot demonstrations, my apologies for the inaccuracy, but my understanding is there aren't.
> >
> > Nevertheless, most of my other questions could be answered. I encourage the authors to incorporate all the new results and ablations into the paper or supplementary.
> >
> > Overall, the ablation on training only the classification head reveals some more insights in how much the synthetic images from KnowData and how much the more flexible fine-tuning contributes to the final results. Some comparisons are much closer when fine-tuning the same amount of parameters as Synthetic. In the end, there are particular datasets where KnowData works well, e.g. DTD and EuroSAT, but it is not universally convincing.

---

> > > ### Author Response · Authors · 2024-12-03
> > >
> > > Thank you for your valuable feedback. We are pleased that we were able to address most of your questions. You are correct that our current work includes only qualitative evaluations comparing results with and without few-shot demonstrations, as the generation quality without few-shot demonstrations is clearly low. We will incorporate quantitative evaluations in the revised manuscript, as per your suggestion.
> > >
> > > Regarding the ablation study on training only the classification head, we would like to note that KnowData consistently outperforms the Synthetic baseline on ViT-B/16, even on the challenging ImageNet out-of-distribution test datasets. We believe this is because knowledge-enhanced data provides more meaningful and diverse information compared to other baseline synthetic data. Moreover, fine-tuning the model with more layers in image encoders will enable it to fully leverage this additional information, as discussed in lines 282-284 of the paper.
> > >
> > >
> > > As the rebuttal period is nearing its end, we would greatly appreciate your feedback on whether our responses and revisions have adequately addressed your concerns. We are also open to further discussions if needed.
> > >
> > > Thank you once again for your time and comments.

---

### Official Review · Reviewer_XFV4 · 2024-11-04

**Soundness:** 3
**Presentation:** 3
**Contribution:** 3
**Rating:** 6
**Confidence:** 3

**Summary:**

This paper proposes a novel method that use structured knowledge in ConceptNet and unstructured knowledge in Wikipedia to augment the synthetic image description with the help of LLMs. The synthetic data contains more informative contents and do not rely on crawling on the web. The resulted text descriptions are then fed to diffusion models to generate synthetic images.

**Strengths:**

1. A novel method to enhance the fine-tuning data of CLIP models
2. Circumvent the usage of crawling.
3. Good downstream task performance improvements

**Weaknesses:**

1. Limited comparison to other synthetic data methods. For example, are the wikipedia retrieval is necessary? Can this part be replaced by a strong LLM, where LLM can serve as a database?
2. A little unfair comparison with baselines. This method unfreezes part of the image encode but other methods only finetune the classification head. This could lead to unfair comparison.

**Questions:**

1. Could you clarify how you unfreeze part of the image encode when fine-tuning? Also can you compare your method with other baselines under the same fine-tuning budget?
2. See Weakness 1
3. Can your method be used to provide pre-training data for CLIP?
4. You have conducted scaling experiments on the image encoder and the fine-tuning dataset. Could you please also test whether your method is applicable to already strong CLIP-based pre-training models, like SynthCLIP [1]
5. Can you compare more fine-tuning baselines using synthetic data, such as LoGoPrompt [2], which shows greater performance.

---

> ### Author Response · Authors · 2024-11-24
>
> Thank you for your valuable suggestions!
>
> >Q1:Could you clarify how you unfreeze part of the image encode when fine-tuning? Also can you compare your method with other baselines under the same fine-tuning budget?
>
> A1: Thank you for your insightful questions. We are happy to provide further clarification.
>
> **Unfreezing Part of the Image Encoder during Fine-tuning:**
>
> As detailed in Appendix G of our paper, we investigated the impact of fine-tuning different portions of the image encoder on model performance. Specifically, Table 9 illustrates that the ViT-B/16 model achieves optimal results when we fine-tune the last 31 layers (which include the classification head) using our 480k ImageNet synthetic dataset. Consequently, for evaluations on the CIFAR, EuroSAT, and ImageNet variant datasets, we chose to fine-tune these last 31 layers. Similarly, for the RN50 model, we fine-tuned the last 44 layers.
>
> The reason for unfreezing these layers is that our knowledge-enhanced synthetic data contains richer and more complex information compared to other baseline synthetic datasets. Fine-tuning only the classification head may not suffice for the model to fully capture and utilize this additional information. By unfreezing more layers, we enable the model to adjust a greater number of parameters, allowing it to learn a better representation of the data distribution and improve overall performance.
>
> **Comparison with Other Baselines Under the Same Fine-tuning Budget:**
>
> In response to your suggestion, we conducted additional experiments where we fine-tuned only the classification head, ensuring a fair comparison with other baselines under the same fine-tuning budget. The results demonstrate that even with this constraint, **KnowData** surpasses the current state-of-the-art methods. Although the accuracy is slightly lower than when more layers are fine-tuned, the performance gain over other baselines underscores the robustness and effectiveness of our approach.
>
> For a detailed comparison of these results, please refer to the following anonymous link: [Classification Head Fine-tuning Results](https://i.ibb.co/x3cZLp3/classification-head-results.png).
>
> We believe these findings highlight the adaptability of our method under different fine-tuning settings and further validate its superiority over existing approaches.
>
> >Q2:Limited comparison to other synthetic data methods. For example, are the wikipedia retrieval is necessary? Can this part be replaced by a strong LLM, where LLM can serve as a database?
>
> A2: Thank you for your insightful questions. We are happy to provide further clarification.
>
> In Table 5, we consider the Pure GPT baseline where we directly prompt GPT-3.5 to generate descriptions about classes (using prompts “write a detailed description about {ci}”). The results show that the Pure GPT baseline performs worse than KnowData which incorporates external knowledge sources. It confirms the importance of external knowledge. So both parts of external knowledge can not be replaced by strong LLM. Wikipedia can provide a strong detailed supplement for ConceptNet knowledge, as shown in Figure 1's examples.
>
> Moreover, a related work, Menon & Vondrick [1], solely utilizes ChatGPT to derive descriptions from class names and directly uses these descriptions to replace class names to obtain text embeddings for classification. It mentions some issues with direct generation by GPT, noting it has shortcomings in factuality. Despite being prompted for visual features of a category, ChatGPT occasionally generates descriptors that, while accurate, reflect other modalities. It lacks diversity, as a word can have multiple meanings, and GPT might only opt to generate using the most common meaning. As Wikipedia retrieval can provide both factuality and diversity, it is important for the whole pipeline.
>
> - [1] Visual classification via description from large language models. ICLR 2023
>
> >Q3:Can your method be used to provide pre-training data for CLIP?
>
> A3: Thank you for your insightful comment. We believe the synthetic data generated by KnowData could serve as a high-quality source of pretraining data for CLIP. While we have not conducted experiments to pretrain CLIP from scratch due to resource constraints, we consider this a promising direction for future work to extend the applicability of our method.

---

> > ### Author Response · Authors · 2024-11-24
> >
> > >Q4:You have conducted scaling experiments on the image encoder and the fine-tuning dataset. Could you please also test whether your method is applicable to already strong CLIP-based pre-training models, like SynthCLIP [1]
> >
> > A4: Thanks for the great suggestion. We add ablation experiments on SynthCLIP-30M based on VitB/16, and compare its performance with OpenAI CLIP. Here are the results:
> >
> > | Method |zeroshot acc on DTD|
> > | :-- | :--: |
> > | OpenAI pretrained VitB/16 | 44.39 |
> > | SynthCI-30M  VitB-16| 19.56 |
> > | OpenAI VitB/16 fine-tuned by KnowData | 57.51 (+13.12) |
> > | SynthCI-30M  VitB/16 fine-tuned by KnowData | 38.83 (+19.27) |
> >
> > It can be seen that KnowData improves the zero-shot performance on SynthCLIP significantly.
> > As shown in the SynthCLIP paper’s Sec 4.2 (Scaling SynthCLIP), while it can have a better performance than CLIP after linear probing, it has a poor zero-shot performance.
> >
> > So SynthCLIP has a larger improvement than CLIP after fine tuning with Knowdata, which proves the usefulness of Knowdata.
> >
> > >Q5: Can you compare more fine-tuning baselines using synthetic data, such as LoGoPrompt [2], which shows greater performance.
> >
> >
> > A5: We thank the reviewer for pointing out the valuable related work.
> >
> > While LoGoPrompt [2] demonstrates impressive performance, it is not directly comparable to our approach due to differences in the problem setting. LoGoPrompt is specifically designed for few-shot scenarios, leveraging real training examples to optimize learnable vectors that represent context words in a prompt. This approach involves selecting examples from the training set for optimization, making it inherently dependent on real data.
> >
> > In contrast, our method operates in a zero-shot setting where only synthetic samples are utilized throughout the entire pipeline, without reliance on any real examples.

---

> > ### Comment · Reviewer_XFV4 · 2024-11-30
> >
> > Thank you for your detailed responses to the concerns raised in my review. I appreciate the additional insights and clarifications provided, which have addressed some of my initial concerns. I am happy to raise the score to 6.

---

> > > ### Author Response · Authors · 2024-12-02
> > > **Response to Reviewer XFV4**
> > >
> > > Thank you again for your valuable feedback and positive rating. Your support is vital to us.

---

### Author Response · Authors · 2024-12-03
**General Response**

We sincerely thank the reviewers for their thoughtful feedback and valuable suggestions. We are pleased that the reviewers recognize our work as a novel method with a neat idea, which circumvents the need for crawling, significantly improves zero-shot performance, and provides a comprehensive set of ablation studies. We also appreciate the positive feedback on the clarity and quality of the writing.

In response to the reviewers' comments, we have conducted additional experiments and analyses to further strengthen our work. Below are the key contributions and updates based on the feedback:
- (XFV4,6gNC)  Comparison with other baselines by fine-tuning only the classification head.   [Results](https://i.ibb.co/x3cZLp3/classification-head-results.png) show that KnowData outperforms baselines even under the same fine-tuning budget, due to knowledge-enabled synthetic data.
- (XFV4)  Comparison to  SynthCLIP-30M based on ViT-B/16. KnowData improves zero-shot performance on SynthCLIP significantly.
- (6gNC) Ablation study on the number of structured knowledge descriptions.
- (oNom) Model improvement in terms of confusion matrix of the class pair.  [Results](https://i.ibb.co/7YCGBHX/cifar100-rn50-confmatrix.png) show that KnowData can effectively reduce the misclassfication in the confused classes.
- (qRxQ) Classes that benefit more from knowledge and diversity-enhanced prompts, as shown by [the top 20 labels with the highest accuracy improvements](https://i.ibb.co/hdWLZX8/ACC-top20-improve-imagenet-vitb16.png) and [the overall LOWESS trends](https://i.ibb.co/fS9KRvr/ACC-improve-imagenet-vitb16.png).
- (qRxQ) Importance of the diversity trick.


We also added additional discussion and analysis, including how domain-specific entity knowledge leads to improvements in VQA and compositional generalization tasks (qRxQ, 6gNC).

Please let us know if you have any further questions. We look forward to continuing the discussion with the reviewers to further improve our paper. Thank you!

---

### Meta-Review · Area_Chair_rh9t · 2024-12-20

**Metareview:**

This paper introduces KnowData, a method that combines structured knowledge from ConceptNet and unstructured knowledge from Wikipedia, to create detailed class descriptions for generating synthesis images. These images are used to finetune vision language models, improving their zero-shot classification performance compared to traditional methods.



There is general appreciation from the reviewers for the ease of understanding. The experimental results highlight the method's advantages, demonstrating improvement in the image classification task. However, several questions and concerns were raised in the initial round of feedback, including: 1) the need for broader testing and justification, both in terms of evaluating datasets (such as iNaturalist) and using more advanced models beyond CLIP, such as more advanced MLLMs; 2) incremental improvement on some datasets; 3) clarification on dataset and fine-tuning procedures; and 4) limited comparison with other methods, as well as concerns regarding the fairness of comparison with the baseline due to differences in fine-tuning approaches. In the rebuttal, the authors successfully address many of the concerns. However, some major issues remain, such as the lack of evaluation on more challenging benchmarks like iNaturalist and the need for the base VL model to extend beyond CLIP to more advanced MLLMs. Given these points, I believe the paper cannot be accepted in its current form and requires major revision.

**Additional Comments On Reviewer Discussion:**

In the initial round of review, a number of questions and concerns were raised, including the need for more justification on harder benchmark datasets and more advanced base models. Additional issues noted were the marginal performance gains on some datasets, unclear details regarding the finetuning steps, fairness of comparisons with baselines, missing ablation studies, and several unclear or inappropriate statements. The authors did provide additional experiments, such as ablation studies on the CLIP score filter-out rate and comparisons between training only a classification head versus training multiple layers, which address concerns about unfair comparisons and some others. The additional clarifications answer some of the questions. However, several major issues remain unresolved, particularly that the base model is still CLIP and that the evaluation datasets should include more challenging options like iNaturalist. Moreover, the quantitative evaluation of how and where the generated prompts outperform normally collected ones is argued by the reviewers not to be the focus of this work.

---

### Decision · Program_Chairs · 2025-01-22

Reject